# The energy savings-oxidative cost trade-off for migratory birds during endurance flight

Scott McWilliams[1]*, Barbara Pierce[2], Andrea Wittenzellner[3], Lillie Langlois[1], Sophia Engel[3], John R Speakman[4,5], Olivia Fatica[2], Kristen DeMoranville[1], Wolfgang Goymann[3], Lisa Trost[3], Amadeusz Bryla[6], Maciej Dzialo[6], Edyta Sadowska[6,7], Ulf Bauchinger[7]

[1]Department of Natural Resources Science, University of Rhode Island, Kingston, United States; [2]Department of Biology, Sacred Heart University, Fairfield, United States; [3]Max Planck Institute for Ornithology, Starnberg, Germany; [4]Institute of Genetics and Developmental Biology, Chinese Academy of Sciences, Beijing, China; [5]Institute of Biological and Environmental Sciences, University of Aberdeen, Scotland, United Kingdom; [6]Institute of Environmental Sciences, Jagiellonian University, Kraków, Poland; [7]Nencki Institute of Experimental Biology PAS, Warszawa, Poland

**Abstract** Elite human and animal athletes must acquire the fuels necessary for extreme feats, but also contend with the oxidative damage associated with peak metabolic performance. Here, we show that a migratory bird with fuel stores composed of more omega-6 polyunsaturated fats (PUFA) expended 11% less energy during long-duration (6 hr) flights with no change in oxidative costs; however, this short-term energy savings came at the long-term cost of higher oxidative damage in the omega-6 PUFA-fed birds. Given that fatty acids are primary fuels, key signaling molecules, the building blocks of cell membranes, and that oxidative damage has long-term consequences for health and ageing, the energy savings-oxidative cost trade-off demonstrated here may be fundamentally important for a wide diversity of organisms on earth.

*For correspondence: srmcwilliams@uri.edu

Competing interests: The authors declare that no competing interests exist.

## Introduction

Endurance exercise challenges the physiology of athletes because high metabolism must be fueled and sustained while avoiding the build-up of metabolites that cause oxidative stress and fatigue. Given that flying is very energetically costly compared to running or swimming (*Butler, 2016*), such a physiological trade-off may be especially acute for migrating birds, among the best high-performance endurance athletes on the planet. During long-duration flights, birds expend energy at a rate >10 times above their basal metabolic rate (BMR) (*Butler, 2016*), whereas among the most extreme and competent human endurance athletes - Tour de France riders - have sustained metabolic rates of around five times BMR (*Hammond and Diamond, 1997*). Furthermore, birds use fats as their primary fuel (about 95%) for high-intensity endurance exercise such as migratory flights (*Jenni and Jenni-Eiermann, 1998*; *Guglielmo, 2010*; *Guglielmo, 2018*), and these fats are highly susceptible to oxidative damage (*Skrip and McWilliams, 2016*). Regulating oxidative balance is important for all air-breathing organisms because reactive pro-oxidant molecules can cause considerable cellular damage and so affect performance, health and potentially longevity (*Halliwell and Gutteridge, 1999*; *Lane, 2005*; *Cooper-Mullin and McWilliams, 2016*). Here, we experimentally demonstrate that the fatty acid composition of fat stores in a migratory bird and especially the acquisition of a

few essential fatty acids, can reduce the energy cost of endurance flight; however, such beneficial energy savings comes at the cost of longer term oxidative damage. This energy savings-oxidative cost trade-off has important implications for the ecology and physiology of birds, as well as potentially for other, non-avian, athletes.

## Results and discussion

One of the more remarkable features of fat metabolism in vertebrates is that in general 'you are what you eat', that is the fatty acid composition of diet has a predominant influence on that of the body-fat stores (*West and Meng, 1968*; *Thomas and George, 1975*; *West and Peyton, 1980*; *Phetteplace and Watkins, 1989*; *Pierce et al., 2004*; *Pierce and McWilliams, 2005*; *Price and Guglielmo, 2009*; *Ben-Hamo et al., 2011*; *Abbott et al., 2012*; *Pierce and McWilliams, 2014*). We took advantage of this feature of fat metabolism and used diet manipulations (see *Table 1*, *Table 2*) to produce European starlings (*Sturnus vulgaris*) with distinct differences in certain essential fatty acids (*Figure 1*).

Such differences in fatty acid composition of fat stores in starlings, specifically the relative amounts of 18:1, 18:2, and 18:3, are also among the primary longer chain fatty acids that compose the fat stores of wild songbirds especially during migration (*Blem, 1990*; *Pierce and McWilliams, 2005*; *Pierce and McWilliams, 2014*). We used these two groups of starlings with different fatty acid composition of their fat stores to directly test the hypothesis that birds with more essential omega-6 and −3 PUFA (18:2 and 18:3) comprising their fat depot have enhanced exercise performance during long-duration flights, as previously shown for short-duration flights of a few minutes

**Table 1.** Ingredients and composition of the two semi-synthetic diets fed to European starlings used in Experiments I and II.

The two diets were isocaloric and composed of 42% carbohydrates, 23% protein, and 20% fat. Different amounts of soybean and olive oil were used to produce two diets that differed only in their fatty acid composition (see *Table 2*).

| | MUFA | | PUFA | |
|---|---|---|---|---|
| Ingredients | % wet mass | % dry mass | % wet mass | % dry mass |
| Glucose* | 16.87 | 39.35 | 16.87 | 39.19 |
| Casein† | 8.23 | 19.20 | 8.23 | 19.12 |
| Cellulose‡ | 2.14 | 4.99 | 2.14 | 4.97 |
| Salt mixture§ | 2.06 | 4.80 | 2.06 | 4.78 |
| Olive oil¶ | 7.82 | 18.24 | 4.11 | 9.60 |
| Soybean oil** | 0.41 | 0.96 | 4.11 | 9.60 |
| Amino acid mix†† | 1.15 | 2.69 | 1.15 | 2.68 |
| Vitamin mix‡‡ | 0.16 | 0.38 | 0.16 | 0.38 |
| Mealworms§§ | 2.65 | 6.19 | 2.65 | 6.16 |
| Agar¶¶ | 1.37 | 3.20 | 1.37 | 3.19 |
| Water | 57.14 | | 57.14 | |

*Glucose, VWR International GmbH, Darmstadt, Germany;.

†Casein, Affymetrix UK Ltd., High Wycombe, UK.

‡Alphacel, MP Biomedicals, Solon, OH, USA.

§Brigg's salt mix, MP Biomedicals, Solon, OH, USA.

¶Tip Native brand Olive oil (glass bottle, Vandemoortele Deutschland GmbH).

**Soya oil, Sojola-brand Soja Oil; Vandemoortele Deutschland GmbH.

††Amino Acid Mix, Sigma-Aldrich, St. Louis, MO, USA.

‡‡AIN-76 vitamin mix, MP Biomedicals, Solon, OH, USA.

§§Freeze-dried mealworms: Futtermittel Hungenberg Brand, Germany; ca. 45% protein, 33% fats, 7% carbohydrates, and 15% indigestible fiber and ash (*Finke, 2002*).

¶¶Agar, Ombilab-laborenzentrum GmbH, Bremen, Germany.

**Table 2.** Fatty acid composition (% ± SE) of the monounsaturated (MUFA) and polyunsaturated (PUFA) fatty acid diets plus mealworms, and of furcular fat from European starlings fed each diet for 4+ months.

Fatty acid concentration was directly measured by gas chromatography in lipids extracted from the diets and the furcular fat.

| Fatty acid* | MUFA | | PUFA | |
|---|---|---|---|---|
| | Diet w/mealworms | Furcular fat (n = 13) | Diet w/mealworms | Furcular fat (n = 16) |
| 16:0 | 12.52 | 16.61 ± 0.81 | 12.73 | 16.63 ± 0.65 |
| 16:1 | 0.00 | 3.34 ± 0.34 | 0.00 | 3.35 ± 0.32 |
| 18:0 | 1.38 | 2.69 ± 0.74 | 1.81 | 3.74 ± 0.56 |
| 18:1[†] | 70.65 | 67.81 ± 1.18 | 51.70 | 56.73 ± 0.90 |
| 18:2[†] | 14.65 | 9.54 ± 1.32 | 29.34 | 17.70 ± 1.17 |
| 18:3[†] | 0.50 | 0.00 ± 0.00 | 4.11 | 1.75 ± 0.34 |

*Fatty acid nomenclature = C:D where C refers to the number of carbon atoms in the chain and D refers to the number of double bonds present. Other fatty acids found in <1% of the lipid portions of diet were 12:0, 14:0, 20:1, 22:6, 24:1.

[†]Fatty acid composition (% ± SE) of furcular fat from birds fed MUFA diets was significantly different from that of birds fed PUFA diets, in 18:1, 18:2, and 18:3 fatty acids. .

(*Pierce et al., 2005*; *Price and Guglielmo, 2009*). After 15 days of ramping up flight-training, we flew starlings (*n* = 33) for 6 hr (±5 min) in a windtunnel set at a fixed speed of 12–12.5 m/s, the equivalent of a ca. 260 km non-stop flight, and used doubly labeled water to measure energy expended during the 6 hr flight (see Materials and methods for details).

PUFA-fed birds composed of more essential omega-6 (18:2) and omega-3 (18:3) PUFA expended 11% less energy compared to MUFA-fed birds during the 6 hr endurance flight (*Figure 2a*; Diet effect: $F_{1,28}$ = 8.88, p=0.006; Body mass covariate: $F_{1,28}$ = 13.75, p=0.001; Diet × Body mass interaction: $F_{1,28}$ = 0.25, p=0.62). As expected from the 11% energy savings, PUFA-fed birds lost substantially less body mass during the 6 hr flight (7.04 ± 0.30%) compared to MUFA-fed bird (9.01 ± 0.41%; *Figure 2—figure supplement 1*). We replicated this experiment with a second group of hand-raised starlings (n = 36) fed the same PUFA or MUFA diets, trained in the same way in the same wind-tunnel, to measure basal metabolic rate (BMR) and oxidative status of flight-trained as well as control, sedentary birds at different time points during exercise training (see Materials and methods, Experiment II). Exercise training, in general, and the transition to the migratory-state for birds, specifically, might result in upregulation of many fundamental aspects of physiology (*McArdle et al., 2010*; *Guglielmo, 2010*; *Piersma and Gils, 2011*) which may require an increase in BMR. However, we found that BMR measured 36–40 hr after the longest flight was similar for MUFA- and PUFA-fed birds (*Figure 2b*) and was not influenced by flight-training (Training x Diet interaction: $F_{1,25}$ = 0.04, p=0.85; Training effect: $F_{1,25}$ = 0.02, p=0.88; Diet effect: $F_{1,25}$ = 0.61, p=0.44).

The 11% energy savings (*Figure 2a*) achieved by starlings composed of more essential omega-6 PUFA during long-duration flights is quite substantial compared to other performance-enhancing contexts for other organisms. For example, mutant strains of *E. coli* bacteria that did not incur the biosynthetic costs of the amino acid tryptophan used 0.01% less energy than wild types (*Dykhuizen, 1978*). Naked mole rats which inhabit the dark world of borrows saved 2% of their energy budgets by not developing a visual system (*Cooper et al., 1993*). Differences in the energy costs of the top-placing elite athletes such as Tour de France riders (*Saris et al., 1989*; *Hammond and Diamond, 1997*; *Santalla et al., 2012*) are on the order of <5%. Thus, avian migratory performance is not only extreme in terms of the amount of energy burned per unit time (*Butler, 2016*), but also in the potential savings of energy during long-duration exercise associated with having their fat stores composed of more essential omega-6 (18:2) and omega-3 (18:3) PUFA.

In theory, selectively eating and hence storing certain long-chain unsaturated fatty acids may be advantageous because such fatty acids (i) may affect composition and key functions of lipid-rich cell membranes (membrane hypothesis), (ii) may be metabolized more quickly (fuel hypothesis), or (iii)

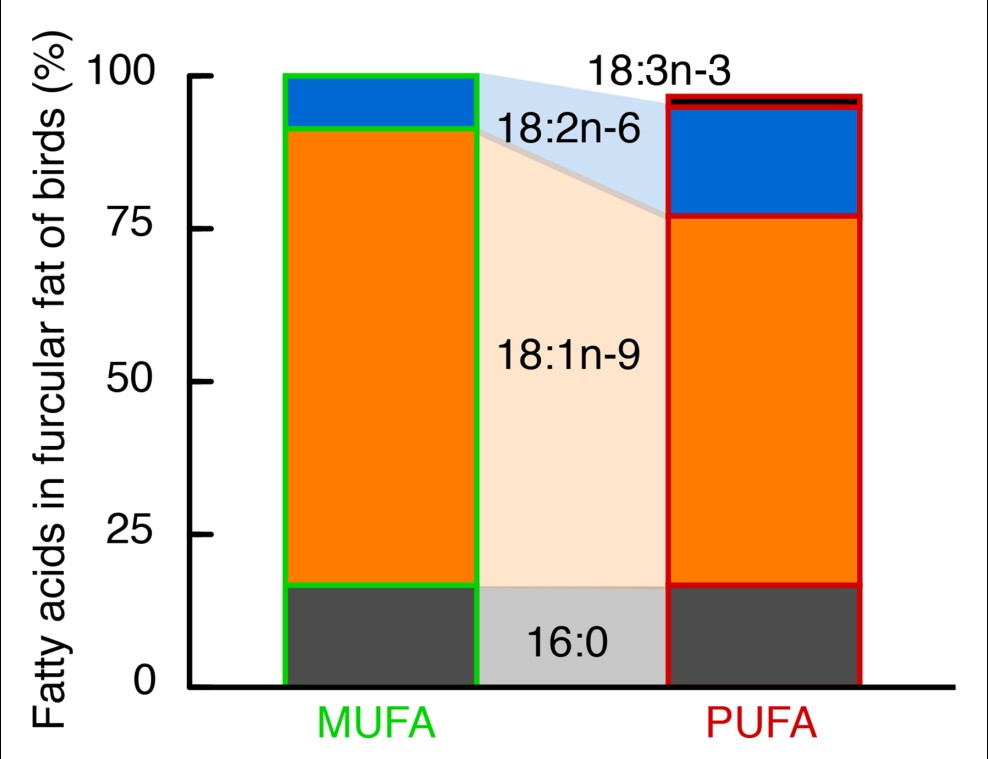

**Figure 1.** Fatty acid composition of stored fat in European starlings was largely determined by their diet. Hand-raised European starlings were fed over 4+ months one of two isocaloric diets (MUFA or PUFA) that differed only in the relative amounts of mono- and polyunsaturated fats (*Table 1*, *Table 2*), specifically the amounts of omega-9 (18:1), and the so-called 'essential' omega-6 (18:2) and omega-3 (18:3) (no. carbons in fatty acid backbone: no. double bonds, and the 'omega' designation identifies the location of the first double bond from the terminal end). The stored fat (in the furcular region) of MUFA-fed birds was 75% 18:1% and 10% 18:2, whereas that of PUFA-fed birds was 60% 18:1% and 20% 18:2 and 18:3. Importantly, these three fatty acids primarily composing the fat stores of our hand-raised starlings (i.e. 16:0, 18:1, 18:2) are the same three fatty acids that predominate in the fat stores of free-living passerines sampled during their migration (*Pierce and McWilliams, 2005*).

may stimulate key facets of aerobic metabolism (signal hypothesis) (*Price, 2010*; *Pierce and McWilliams, 2014*). In rats and humans, high levels of essential omega-6 PUFA in muscle membrane phospholipids have been associated with improved endurance capacity (*Ayre and Hulbert, 1996*; *Ayre and Hulbert, 1997*; *Andersson et al., 1998*). Maximum running speed, a short-term performance measure, in 30 species of mammals was positively correlated with the omega-6 fatty acid content in skeletal muscle phospholipids (*Arnold et al., 2015*). Our own recent work and that of colleagues confirmed that two different passerines (Red-eyed vireos [*Vireo olivaceous*], White-throated sparrows [*Zonotrichia albicollis*]) with fat stores composed of more omega-6 PUFA (18:2) have improved performance during short-term intense exercise (*Pierce and McWilliams, 2005*; *Pierce et al., 2005*; *Price, 2010*). Yellow-rumped warblers (*Setophaga coronata*) with enriched omega-3 PUFA in their muscle phospholipids decreased muscle oxidative enzymes, although these changes in muscle metabolism were not associated with changes in flight performance (*Dick and Guglielmo, 2019a*). Unlike most of these studies that were largely mensurative or measured exercise performance over shorter flights (mostly 20–30 min), our experimental results here demonstrate that migratory birds with fat stores composed of more essential omega-6 PUFA had improved performance (i.e. less energy used per km) during endurance flights (6 hr, ca. 260 km) compared to birds composed of more MUFA, and a recent companion study (*Carter et al., 2020*) suggests that this is because of the signaling properties of omega-6 PUFA.

We also tested the hypothesis that this enhanced performance comes with metabolic costs. Such enhanced endurance performance is associated with more lipid peroxide production which must be quenched by upregulation of the endogenous antioxidant system or, if not adequately quenched,

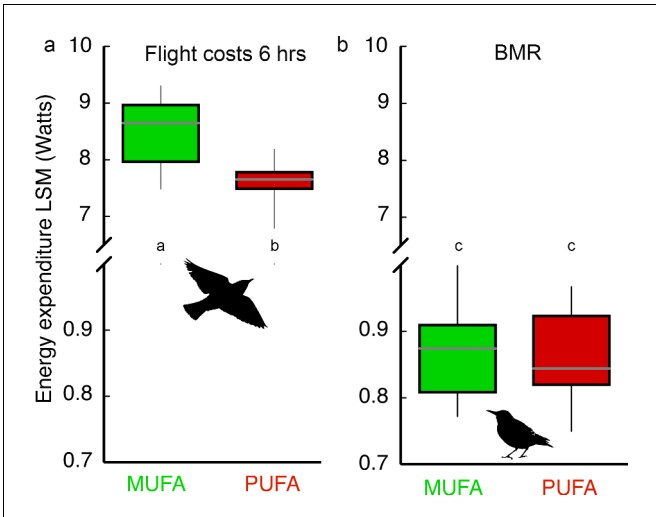

**Figure 2.** Dietary fatty acid manipulation affects flight costs but not costs of self-maintenance. Box plots (mean, 5% and 95% CI, range) for (**a**) Energy expenditure (measured with doubly labeled water) during a 6 hr ca. 260 km flight in a windtunnel for starlings (n = 33) fed one of two diets (n = 16, 17) and after 15 days of flight-training. Body mass was used as a covariate in the ANCOVA comparison of diet effects on energy expenditure, and (**b**) Basal metabolic rate (BMR; measured using open-flow respirometry) of MUFA- (n = 19) and PUFA-fed (n = 17) European starlings that were flight-trained or sedentary. Means with different letters within each panel are significantly different (p<0.05).

The online version of this article includes the following figure supplement(s) for figure 2:

**Figure supplement 1.** Change in (a) body mass, (b) Beta-hydroxbutyrate, and (c) uric acid in plasma of European starlings (*n* = 33) during their long-duration (6 hr) flights in Experiment I.

would increase oxidative damage. We measured plasma indicators of antioxidant status in flight-trained as well as sedentary (untrained) starlings at three different time points during training: before the start of flight training in the windtunnel ('Pre-training'), immediately after a long-duration flight on Day 15 ('Post-flight'), and 1.5 days afterwards ('Recovery').

Flight-training over more than 2 weeks did not affect baseline levels of oxidative damage (compare Pre-training and Recovery; *Figure 3c*) while antioxidant capacity decreased in flight-trained but not sedentary birds (*Figure 3b*) and plasma uric acid decreased over time in both trained and sedentary starlings (*Figure 3a*; see *Table 3* for detailed statistical results). A long-duration flight on Day 15 (average flight times for PUFA: 161.6 ± 31.2 min, and MUFA: 180.2 ± 28.3 min) was associated with increased plasma uric acid (*Figure 3a*), a product of protein metabolism in birds and also a potent antioxidant (*Sautin and Johnson, 2008*), and a reduction in plasma triglycerides (*Figure 3—figure supplement 1*). Regardless of flight-training, birds fed more essential omega-6 PUFA had higher oxidative damage than MUFA-fed birds (main effect of flight; *Figure 3c*) suggesting a fundamental oxidative cost of being composed of more polyunsaturated fatty acids.

Increased energy metabolism during exercise is often associated with increased production of pro-oxidants regardless of the fuel types used (i.e. carbohydrates, protein, fats) which causes oxidative damage if not quickly quenched by dietary antioxidants and/or by increased production of antioxidant enzymes (e.g. superoxide dismutase, glutathione peroxidase) (*Halliwell and Gutteridge, 1999 Costantini et al., 2007*; *Costantini, 2008*; *Costantini et al., 2008*; *Monaghan et al., 2009*; *Jenni-Eiermann et al., 2014*; *Dick and Guglielmo, 2019b*). Animals performing exercise such as birds during a long-duration flight, and those that rely on large amounts of fat as fuel such as migratory birds, are especially prone to oxidative stress because (1) an increase in activity may increase production of ROS in excess of the immediate capacity of antioxidants to quench them, and (2) stored and structural fats, particularly PUFAs, are especially vulnerable to oxidative damage because of their chemical structure, and may generate their own variety of pro-oxidants (*Cooper-Mullin and McWilliams, 2016*; *Skrip and McWilliams, 2016*). Importantly, we found no significant change in oxidative damage or oxidative capacity immediately after flights (compare Post-flight to Pre-training

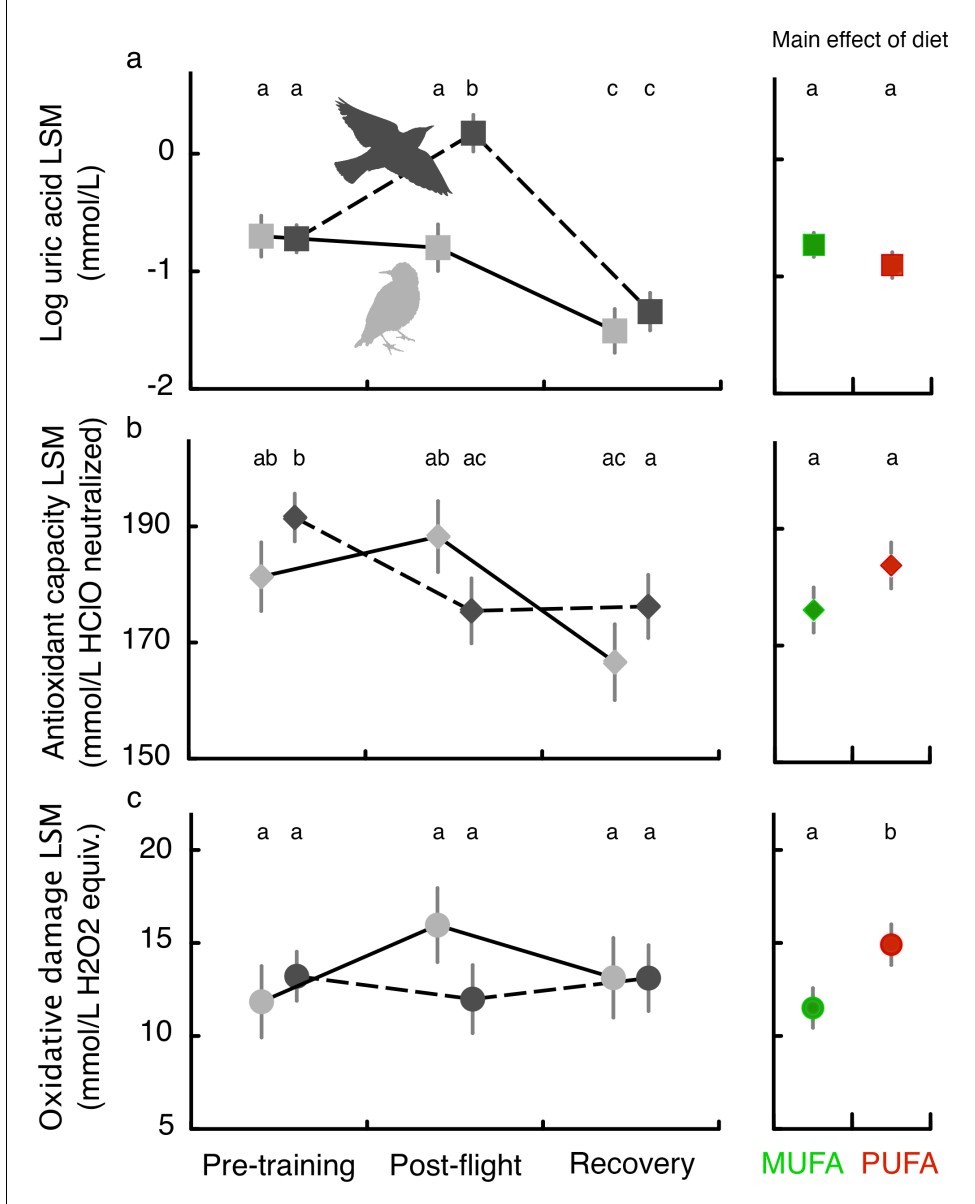

**Figure 3.** Oxidative status of European starlings associated with flight-training and in relation to diet quality. Oxidative status (see Materials and methods, Experiment II) of MUFA-fed or PUFA-fed European starlings was measured in blood plasma at three different time points for flight-trained (black symbols and dashed lines) and untrained, sedentary control (gray symbols and lines) birds: before the start of flight training in the windtunnel (Pre-training), immediately after a long-duration flight on Day 15 (Post-flight), and ca. 1.5 days afterwards (Recovery). Untrained sedentary birds were sampled on the same days but were never exposed to flight-training. Body mass and date of measurement were included as a fixed covariate so we report the results as least square means (LSM). The comparison of Recovery and Post-flight timepoints reveals the effect of the long-duration flight: birds had very similar total flight times over the 15 days of exercise training with the primary difference being whether we sampled the bird's blood immediately after flight (Post-flight) or 2 days after their final flight (Recovery). The main effect of diet (MUFA vs. PUFA) on oxidative status is shown in the right panels. Means with different letters across the three timepoints, or for the main effect of diet, are significantly different (p<0.05). The online version of this article includes the following figure supplement(s) for figure 3:

**Figure supplement 1.** Plasma metabolites of European starlings associated with flight-training and in relation to diet quality.

**Table 3.** The effect of flight (flight-trained for 15 days in windtunnel or not; Trained or Sedentary), diet (MUFA or PUFA), and time (blood sampled at three different time points: before the start of flight training in the windtunnel ('Pre-training'), immediately after a long-duration flight on Day 15 ('Post-flight'), and 1.5 days afterwards ('Recovery')) on plasma metabolites and oxidative status in European starlings in Experiment II. Individuals that did not undergo flight training (i.e. control 'sedentary' birds) were sampled on the same days as flight-trained birds in their same cohort. Test statistics: F-value with denominator degrees of freedom (ddf) and significance level p-value for main factors and their interactions from the linear mixed models.

| Marker | Flight $F_{ddf}$ | p | Diet $F_{ddf}$ | p | Time point $F_{ddf}$ | p | Flight × Diet $F_{ddf}$ | p | Time × Flight $F_{ddf}$ | p | Time × Diet $F_{ddf}$ | p | Diet × Flight × Time $F_{ddf}$ | p |
|---|---|---|---|---|---|---|---|---|---|---|---|---|---|---|
| β-Hydroxybutyrate | $4.72_{32.9}$ | **0.04** | $14.0_{32.9}$ | **<0.001** | $14.0_{62.3}$ | **<0.001** | $0.26_{32.9}$ | 0.61 | $9.44_{62.3}$ | **<0.001** | $0.52_{62.3}$ | 0.60 | $0.32_{62.3}$ | 0.73 |
| Total triglycerydes | $0.32_{32.6}$ | 0.57 | $0.13_{32.6}$ | 0.72 | $9.62_{60.0}$ | **<0.001** | $0.22_{32.6}$ | 0.88 | $6.90_{60.0}$ | **0.002** | $0.40_{60.0}$ | 0.67 | $0.26_{60.0}$ | 0.97 |
| Uric acid | $9.12_{33.5}$ | **0.004** | $0.65_{33.5}$ | 0.43 | $25.94_{63.5}$ | **<0.001** | $1.44_{33.5}$ | 0.24 | $4.32_{62.5}$ | **0.02** | $0.34_{62.5}$ | 0.71 | $0.42_{62.5}$ | 0.66 |
| Antioxidant capacity (Oxy adsorbent assay) | $1.05_{31.5}$ | 0.31 | $1.72_{31.5}$ | 0.20 | $6.00_{58.9}$ | **0.004** | $1.9_{31.5}$ | 0.18 | $4.34_{58.9}$ | 0.17 | $0.90_{58.8}$ | 0.41 | $0.36_{58.8}$ | 0.70 |
| Oxidative damage (dROM assay) | $0.26_{33.1}$ | 0.61 | $5.71_{33.1}$ | **0.02** | $0.27_{62.0}$ | 0.77 | $1.86_{33.1}$ | 0.18 | $1.2_{62.0}$ | 0.31 | $1.50_{62.0}$ | 0.23 | $0.27_{62.0}$ | 0.80 |

and Recovery in *Figure 3*) suggesting that birds were capable of contending with the oxidative costs associated with a given flight. Instead, migratory birds pay the oxidative costs of being composed of more polyunsaturated fatty acids (primarily 18:2 and 18:3) over the long-term (e.g. migration period of the annual cycle) while gaining some energy savings only during a given migratory flight.

Collectively, our study provides compelling evidence that avian athletes – just like human athletes (*Mickleborough, 2013*; *Neubauer and Yfanti, 2015*; *Pingitore et al., 2015*) – face considerable trade-offs when deciding what to eat to enhance their performance. Given that fat composition of diet largely determines that of fat stores, migratory birds that consume diets with more essential omega-6 PUFA (18:2) can substantially enhance their exercise performance (i.e. expend less energy) during long-duration flights without incurring increased self-maintenance (BMR) costs. However, this enhanced performance during flights has longer-term costs in terms of increased oxidative damage when fat stores are composed of more PUFA. Cafeteria-style diet choice experiments with migratory songbirds have shown that birds carefully discriminate between diets that differ only in their fatty acid composition (*Pierce et al., 2004*; *Pierce and McWilliams, 2014*), and on average prefer to consume a composite diet that contains a 1:2 ratio of 18:2 to 18:1 (*Pierce and McWilliams, 2014*). This indicates that birds may choose among diets to optimize the trade-off between enhanced flight performance (more 18:2), while reducing the long-term costs of being composed of more long-chain PUFA. Migratory birds can also optimize this energy savings-oxidative cost trade-off by being composed of more n-3 and/or n-6 PUFAs only during migration periods when energy demands and fat catabolism are most extreme, and then become more monounsaturated in composition during non-migration periods - such seasonal changes in fatty acid composition are commonly observed in migratory birds (*Pierce and McWilliams, 2005*; *Santalla et al., 2012*; *Pierce and McWilliams, 2014*). Such a trade-off may become especially detrimental if foods with different quantities of micronutrients (notably long-chain PUFAs and antioxidants) are not available in nature, in which case birds may be unable to ameliorate such a trade-off through careful choices of diet. Given the long-term consequences of oxidative damage for health and aging (*Halliwell and Gutteridge, 1999*), and the enhanced exercise performance associated with certain fatty acids that incur long-term oxidative costs (i.e. the energy savings-oxidative cost trade-off demonstrated here), strong evolutionary forces likely act on these diet choices for human and non-human vertebrates especially those that at times require enhanced exercise performance.

## Materials and methods

### Study species and overall experimental plan

We used hand-raised European starlings (*Sturnus vulgaris*) from southern Germany (n = 95 total) for this two-part study because starlings are common migratory birds in western Europe (*Feare, 1984*)

and they have been successfully hand-raised and trained to fly in windtunnels (*Engel et al., 2006a*). European starlings from this southeastern German population are medium-distance diurnal migrants that leave for wintering grounds in October and November and return to their Bavarian breeding grounds in April (*Bairlein, 2014*). Migratory distances for this population vary from many hundreds of kilometers to several thousand kilometers, with some individuals overwintering in the Euro-Mediterranean region and others in northwest Africa (*Bairlein, 2014*). European starlings are also quite social and curious, and quickly learn to successfully fly together in a given windtunnel as demonstrated by several recent studies (*Carter et al., 2020*; *Casagrande et al., 2020*). We conducted two complimentary experiments that involved feeding starlings over many months two diets that differed only in the relative amounts of mono- and polyunsaturated fats (*Table 1*), specifically the amounts of omega-9 18:1 and omega-6 (18:2) and omega-3 (18:3) (no. carbons in fatty acid backbone: no. double bonds). Previous work has established that these are the primary longer chain fatty acids in wild songbirds especially during migration (*Blem, 1990*; *Pierce and McWilliams, 2005*; *Pierce and McWilliams, 2014*), and that the fatty acid composition of birds reflects that of their diet (*Pierce and McWilliams, 2005*; *Pierce et al., 2005*; *Price and Guglielmo, 2009*) which we also demonstrate here (*Table 2*).

All procedures adhered to the ethical guidelines of the North American Ornithological Council (*Fair et al., 2010*) and were approved by the University of Rhode Island IACUC (Protocols #AN09-09-009, AN08-02-014) and the Government of Upper Bavaria, Germany (AZ 55.2-1-54-2532-216-2014).

## Starling care, aviaries, and experimental diets
### Capture and maintenance of birds
We collected 40–45 nestling (5–11 day-old) European starlings from nest boxes in late-April to early-May 2005 and 2015 from a colony in Upper Bavaria, South Germany (47°58' N, 11°13' E). We hand-

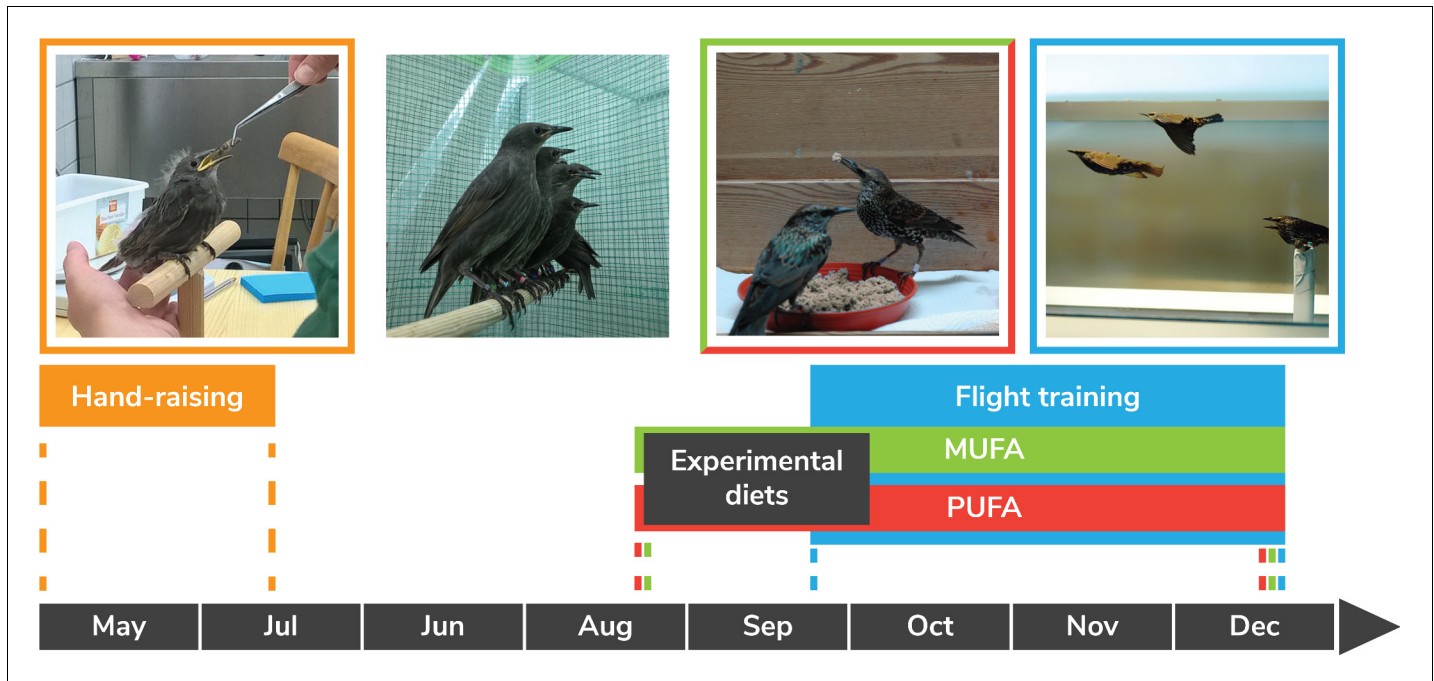

**Figure 4.** Experimental design for both Experiments I and II from hand-raising of nestling European starlings, to acclimation to one of two experimental diets (both composed of 42% carbohydrates, 23% protein, and 20% fat but differing in the amount of polyunsaturated (PUFA) or monounsaturated (MUFA) fatty acids), and then to flight training in a windtunnel. During fall, cohort groups of 2–3 starlings were flight trained in the windtunnel for 14 days and then flew on Day 15 a long-duration (usually 6 hr) flight (see *Figure 5*) during which energy expenditure and plasma indicators of metabolism and oxidative status were measured. For Experiment I, we flight-trained 36 starlings of which 33 completed their 6 hr long-duration flight (16 fed the MUFA diet, 17 fed the PUFA diet). For Experiment II, we added 1–2 untrained, sedentary (control) starlings in each training cohort which resulted in a total of 19 MUFA-fed birds (11 flight-trained, eight sedentary) and 17 PUFA-fed birds (nine flight-trained, eight sedentary).

raised the chicks at the Max Planck Institute for Ornithology (MPIO), Seewiesen, Germany (*Figure 4*) on a high-protein diet of bee larvae, crickets, and beef heart with vitamin mixture until they were able to feed independently (ca. 35 days old). During this hand-raising, small groups (2-5 ) of hatchlings of similar age were housed in wooden boxes (25 × 25 × 25 cm high) lined with paper towels that were changed after every feeding. This same diet has been successfully used to raise starlings for windtunnel experiments (*Engel et al., 2006b*; *Schmidt-Wellenburg et al., 2008*) and growth rates, molt, and size at fledging for our starlings were similar to previous studies (*Engel et al., 2006a*). At ca. 35 days old, fledglings were able to feed independently and so they were housed in groups of 10–15 individuals per aviary (1.5 × 2 × 2.5 m) and offered ad libitum live mealworms, fresh fruits, and vegetables. At the age of ca. 50 days, groups of 10–15 starlings were then moved to larger aviaries (2 × 4 × 2 m) and offered fresh fruit and vegetables, and a mixed diet of beef heart, yogurt, rusk flour, eggs, mealworms, and other insects (e.g. crickets, bee larvae, wax worms, green bottle fly larvae), soy oil, nuts and dried fruits, and vitamins and minerals.

We randomly assigned 70–75 day old starlings to one of two semi-synthetic agar-based diets (*Murphy and King, 1982*) that were isocaloric and composed of 42% carbohydrates, 23% protein, and 20% fat (*Table 1*). The two diets (hereafter, MUFA, PUFA) were identical in terms of

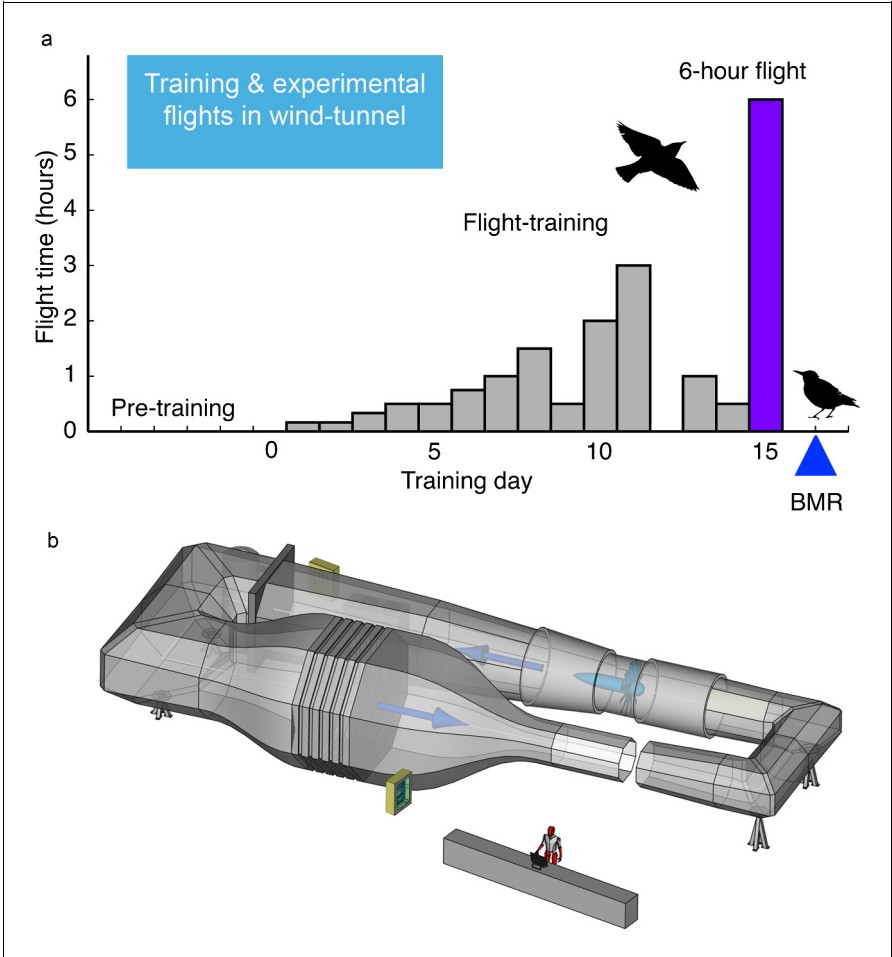

**Figure 5.** Fifteen-day flight-training schedule used for European starlings that were flown in the Max Planck Institute for Ornithology (MPIO) windtunnel. (**a**) Amount of flying time each day was increased until the final 6 hr flight on Day 15; BMR was measured overnight on the day after this longest flight (**b**) Schematic of the MPIO windtunnel showing the 2 × 1.5 m working section (closest to the person) that the birds were trained to fly into through the gap (which was then closed during flights). A large fan, shown directly across from the working section, created the wind which was made laminar with a series of screens just before the compression of the tunnel (bottom left). The wind velocity was accelerated with the 20:1 compression of the tunnel just before the working section.

macronutrient composition and differed only in the relative amounts of certain mono- and polyunsaturated fats (*Table 2*). The macronutrient composition of the two semi-synthetic diets simulates a natural high-lipid fruit diet (*Johnson et al., 1985*; *Smith et al., 2007a*), and the primary fatty acids in the diet (>90% 16:0, 18:1, and 18:2) are also the most common fatty acids in natural fruits (*Pierce and McWilliams, 2014*) and in songbirds that eat fruits during migration (*Pierce and McWilliams, 2005*; *Pierce and McWilliams, 2014*). European starlings are like many fall-migrating songbirds in that they switch to eating largely fruits (*Feare, 1984*), much to the chagrin of many vineyard owners. For Experiment I, the random assignment resulted in roughly equal numbers of males and females fed each of the two diets, whereas for Experiment II we randomly assigned only males to the two diets because the females were used for a different experiment.

## Light cycles

The light schedule from May to December 2005 and 2016 was the same as natural day:night photoperiod at the latitude of Seewiesen, Germany. Large windows ensured exposure to natural light levels, although there were additional supplemental lights inside the aviaries on the same light cycle (bulbs were Osram LUMILUX T8 58 W/865). We did not directly verify that such decreases in light levels in fall induced starlings to increase food intake, although many other studies provide such evidence in migratory birds (*Gwinner, 1996*; *Helm et al., 2009*; *Bulte and Bairlein, 2013*), or to increase *Zugunruhe* (nocturnal activity), since starlings are diurnal migrants.

## Flight training schedule

After starlings were fully acclimated to their respective diets (1+ months), we then flew starlings for prescribed amounts of time each day for 15 days in a recirculating windtunnel designed for studying bird flight at the Max Planck Institute for Ornithology (MPIO), Seewiesen, Germany (*Schmidt-Wellenburg et al., 2007*, *Figure 5*). During the flight training period during fall (Sept-Dec), every 3 days we randomly selected two to threeindividuals from a given diet group to start the ca. 2-week flight training, although this flight-group selection was stratified by extent of Pre-basic I molt (*Ginn and Melville, 1983*) - birds furthest along in molt were flight-trained earliest. This stratified random sampling of individuals ensured that all birds had completed their flight feather molt prior to flight training (i.e. molt scores were >70 in all cases based on Ginn and Mellville [*Ginn and Melville, 1983*]) and the 20 groups of starlings completed their flight training by early December. For Experiment II, we also randomly selected a subset of starlings fed each diet that were not exposed to the flight-training - hereafter, the untrained, sedentary 'control' group. While flight-trained birds were being trained each day to fly in the windtunnel, their companion untrained, sedentary control birds remained in their aviaries adjacent to the windtunnel. On Day 15, while the trained birds were flying for up to 6 hr, the sedentary birds were kept in individual cloth cages without access to food and water.

The flight-training schedule and the duration of the longest (6 hr) flight were chosen based on logistics as well as ecological relevance. The 15-day flight-training schedule and flying conditions (always 12 m s$^{-1}$ wind speed and 15°C) had been used in previously successful experiments at MPIO designed to fly barn swallows (*Hirundo rustica*) and starlings (*Sturnus vulgaris*, *Sturnus roseus*) for long durations (*Engel et al., 2006a*; *Engel et al., 2006b*; *Engel et al., 2006c*; *Schmidt-Wellenburg et al., 2008*; *Schmidt-Wellenburg et al., 2007*), and free-living songbirds including starlings typically complete their migration from breeding to wintering areas over many days of flying and stopovers (*Feare, 1984*; *Newton, 2006*). The duration of the final longest flight (6 hr, 260 km) was sufficiently long to provide adequate turnover of the isotope-labeled water and so ensure accurate measurements of energy expenditure using the doubly labeled water (DLW) technique (*Speakman, 1997*). The 6 hr duration of flight is also within the range of typical single-day migratory flights for many free-living songbirds including starlings (*Feare, 1984*; *Newton, 2006*). During flight-training, each group of two to three individuals was transferred to smaller aviaries (2 × 0.7 × 2 m) that surrounded the working section of the windtunnel (2 × 1.5 m octagon). During the four pre-training days before the onset of flight training (=day 1), all birds were habituated to the windtunnel; each day they spent up to 30 min perched and taking short flights in the wind-tunnel. We used a specifically experienced 'trainer-bird' during this initial 4-day pre-training period in order to provide guidance to the inexperienced birds. Pre-training time was not included in the total flight time. For

Experiment I, flight-training for each group of three birds involved 15 consecutive days of specified amounts of flying in the windtunnel set to 12 ms$^{-1}$ wind speed and 15°C (day 1 flight time = 10 min, day 2 = 10 min, day 3 = 20 min, day 4 = 30 min, day 5 = 30 min, day 6 = 90 min, day 7 = 60 min, day 8 = 90 min, day 9 = 30 min, day 10 = 120 min, day 11 = 180 min, day 12 = no flight training, day 13 = 60 min, day 14 = 30 min, day 15 = 360 min (*Figure 5*)). The same flight training schedule was used for Experiment II except for logistical reasons birds flew for 20 min on days 1 and 2, for 60 and 30 min on days 5 and 6, respectively, and we added a rest day (no flight training) on day 9. This resulted in a total flight-training time, excluding the longest flight on day 15, of 715 min and 720 min for Experiments I and II, respectively. For each bird, we recorded actual flight time each day and summed the total time spent flying for the entire 15-day period.

## Doubly labeled water to measure energy expenditure during long-duration flight

We used the DLW technique (*Speakman, 1997*) in Experiment I to test the hypothesis that energy expenditure of birds during the long-duration (6 hr) flight was affected by the fatty acid composition of their fat stores. Food was removed from aviaries at lights off on Day 14. On Day 15, the three birds in a given flight-training cohort were captured from their small aviaries 15 min after lights on. At 5 min intervals thereafter, we measured body mass of each of the three birds and then intraperitoneally injected 0.2 µL DLW (BD 0.3 ml Micro-Fine U-100 Sterile Insulin syringe). Amount injected was estimated by differential weighing of the syringe to the nearest 0.0001 g using an analytical Sartorious balance. Following injection birds were kept in a small cloth cage with a perch but no water or food. Precisely 60 min after injection, we collected two 40 µL blood samples into non-heparinized capillary tubes from the jugular vein or after puncture of the brachial vein with a 27G needle. All blood samples in capillary tubes were flame sealed immediately upon collection and stored at room temperature until analysis. Birds were allowed to rest in the windtunnel aviaries for about 5 min before starting their 6 hr flight in the windtunnel set to 12 m s$^{-1}$ wind speed and 15°C. Exactly 6 hr later, and in the same order of injections, birds were removed from the windtunnel mid-flight and bled again. We collected two 40 µL blood samples into heparinized capillary tubes from the brachial vein after puncture with a 17G needle, plus an additional 160 µL blood sample transferred to 0.5 mL Eppendorf tubes and centrifuged at 10,000 RPM. Separated plasma was stored at −80°C until subsequent plasma metabolite measures.

Daily energy expenditure (DEE in watts, J s$^{-1}$) was measured using the DLW technique (*Lifson, 1966*; *Butler et al., 2004*). This method has been previously validated by comparison to indirect calorimetry in a range of small mammals (*Speakman and Król, 2005*). To estimate the background isotope enrichments of $^2$H and $^{18}$O, uninjected birds were weighed (±0.1 g Kern pocket balance) and a 20 µL blood sample was obtained from the jugular vein (*Speakman and Racey, 1987*). Blood samples were immediately heat sealed into 2 × 75 µL glass capillaries, which were stored at room temperature. Analysis of the isotopic enrichment of all blood samples was performed blind, using a Liquid Isotope Water Analyser (Los Gatos Research, USA) (*Berman et al., 2012*). Initially, the blood encapsulated in capillaries was vacuum distilled (*Nagy, 1983*), and the resulting distillate was used for analysis. Samples were run alongside five lab standards for each isotope and International standards to correct delta values to ppm. A single-pool model was used to calculate rates of $CO_2$ production as recommended for use in animals less than 5 kg in body mass (*Speakman, 1993*). There are several approaches for the treatment of evaporative water loss in the calculation (*Visser and Schekkerman, 1999*). We detected no differences in body water pool before and after 6 hr flights. We assumed evaporation of 25% of the water flux (Equation 7.17 in *Speakman, 1997*), which minimizes error in a range of conditions (*Visser and Schekkerman, 1999*; *van Trigt et al., 2001*). Given that our experimental design involved feeding birds diets that were identical in macronutrient composition and differed only in their fatty acid composition, the dietary substrates (carbohydrates, fats, proteins) available as fuels were also identical. Empirically measured variation in the amount of stored fat or protein used as fuel (as reflected in the RQ) results in errors in estimates of energy expenditure not exceeding ±2% (*Black et al., 1986*; *Schmidt-Wellenburg et al., 2008*; *Westerterp, 2017*).

## Basal metabolic rate

We used open-flow respirometry (*Lighton, 2019*) in Experiment II to test the hypothesis that BMR of flight-trained birds was affected by the fatty acid composition of their fat stores. We measured BMR during the night on Day 16 after the birds had recovered from their last and longest-duration flight ca. 1.5 days earlier. BMR was measured with a multi-channel open flow respiratory system. Starlings were weighed and placed in separate plastic respirometric chambers (800 ml) without water or food. All the chambers were additionally covered with dark non-transparent paper and were placed in a Peltier effect temperature-controlled portable cabinet (Sable system International, USA) at 25°C, within the thermoneutral temperature range for starlings (*Lustick and Adams, 1977*; *Geluso and Hayes, 1999*). Sufficient mixing of air in the chamber was achieved by plumbing the air intake in the lower part of the chamber and the outtake in the top part of the chamber. A fresh sample of intake air at standard pressure and room temperature was dried with silica gel driers and divided into two streams (flush and master flow) controlled by Intelligent Multiplexer V5 (Sable system International, USA). The flush air stream was pumped with a Mini Laboport Vacuum Pump, (KNF, Germany) at an air flow rate of 780 ml min$^{-1}$ for each chamber (Brooks mass flow controller, USA). The master air stream was pumped (Vebreclerwerk, Germany) at an air flow rate of 1000 mL min$^{-1}$ (Sierra Instruments mass flow controller, USA) with a mass flow controller V1.1 (Sable system International, USA).

Subsamples of air were collected from the chamber at rate of 150 mL min$^{-1}$. Before passing the $CO_2$ and $O_2$ analyzers the subsample was pre-dried with PC-1 non-chemical drier (Sable system International, USA) and then chemically dried with a magnesium perchlorate (Anhydrone, J.T. Baker, USA) column. Samples were analyzed with a FOXBOX (Sable System International, USA) connected to a computer through UI3 (Sable system International, USA) and data were recorded continuously with ExpeData software (Sable System International, US). Dried samples of air were taken sequentially from the measurement chambers and the reference every 18 min. In each cycle, each measurement chamber was active for 110–120 s. We used data from the last 6 hr of measurement (ca. 4 hr after starting measurements) to be sure that metabolic rate was measured in the post-absorptive state. Rates of $O_2$ consumption and $CO_2$ production were calculated from the values recorded in the last 20 s before switching channels, and the mean of the two lowest recordings were used for further calculations. BMR in watts was calculated based on respirometric equations from *Lighton, 2019*. Because of failure of the $O_2$ analyzer for 21 out of 53 available measurements, we calculated $O_2$ consumption for all birds based on (a) measured $CO_2$ values, and (b) the estimated average RQ of 0.789 for the 32 birds with complete measurements.

## Blood sampling and analyses: antioxidant status and plasma metabolites

At each of the three time points (Pre-flight and Post-flight on Day 15) in both Experiments I and II and just before sacrifice on Day 17 in Experiment II, within 3 min of bird capture we drew ca. 200 μL blood from the brachial vein after puncture with a 27 G needle and collected it in heparinized capillary tubes. All birds were sampled at each time point while fasted for at least 12 hr (i.e. after an overnight without food and before offered food on the Pre-training and Recovery days, and after an overnight without food plus their longest flight on the Post-flight day). Within 10 min of sampling, blood samples were centrifuged at 214 *g* for 5 min to separate plasma from the RBCs. Plasma was stored at −80°C until lab analyses of antioxidant status (i.e. antioxidant capacity, oxidative damage) and plasma metabolites (beta-hydroxybutyrate, triglycerides, and uric acid) were conducted.

Non-enzymatic antioxidant capacity was measured with the OXY-adsorbent test (OXY) in the plasma diluted 1:100 dH$_2$O (concentration unit = mmol L$^{-1}$ of HClO neutralized; Diacron International, Grosseto, Italy). OXY directly measures the ability of a plasma sample to neutralize the oxidant hypochlorous acid and provides an index of non-enzymatic antioxidant capacity, without being complicated by the interaction of uric acid (*Alan and McWilliams, 2013*; *Skrip and McWilliams, 2016*; *Costantini, 2016*). Oxidative damage was measured with the d-ROMs test in plasma diluted 1:3 0.9%NaCl (concentration unit = mmol L$^{-1}$ H$_2$O$_2$ equivalents; Diacron International). ROMs measured in this test are primarily hydroperoxides, which are produced when ROS interact with many different biological macromolecules (*Costantini, 2016*), but in plasma, are primarily produced during lipid oxidation events (*Davies, 2016*; *Ito et al., 2017*).

Total triglyceride (TRIG) is a marker of fatty acid anabolism and was measured by sequential endpoint assay (Sigma, St. Louis, Missouri; 5 µL plasma, 240 µL reagent A, 60 µL reagent B) by first measuring free glycerol and then subtracting free glycerol concentration from measured total triglyceride concentration. Beta-hydroxybutyrate is a marker of fatty acid catabolism and was measured by kinetic assay (Cayman Chemical Assay; 5 µL plasma, 50 µL of developer solution which was a mix of 2.4 mL enzyme solution and 100 µL colorimetric detector). Uric acid is a product of protein catabolism and was measured by endpoint assay (TECO Diagnostics, Anaheim, California; 5 µL, sample, 300 µL reagent) using a modified protocol from *Smith et al., 2007b*.

## Fatty acid composition of diet and furcular fat

Fatty acid composition of the two semi-synthetic diets (*Table 2*) was measured by gas chromatography in lipids extracted using a modified Folch method (*Carter et al., 2020*) from composited subsamples of each diet taken over the course of the experiments. We collected adipose tissue from the visible fat stores in the furcular area of starlings used in Experiment I (*n* = 29) 5–7 days after their longest flight in the windtunnel. We conducted the biopsies of this furcular fat using the methods outlined in *Rocha et al., 2016*. Briefly, upper breast feathers were wetted and moved aside to expose the skin and visible subcutaneous yellow-colored fat stores. We selected an area without visible capillaries, disinfected the area with antiseptic solution, and applied a topical anaesthetic gel to this area. After ca. 10 min, we then pinched the skin in this area to ensure no pain response by the bird. We then made an ca. 3 mm-long incision in the skin using a sharp scalpel, pulled a small piece of adipose tissue (ca. 10–20 mg) through the incision using sterile forceps, and cut the sample under the forceps using sterile scissors. The incision area was then realigned and a thin layer of veterinary tissue glue was applied to seal the incision. All birds were checked weekly thereafter and the wound was completely healed within 2–3 weeks.

Dietary fat and furcular fat samples were stored at −80°C until fatty acid composition was measured by gas chromatography. We thawed samples, extracted total lipids using a modified Folch method (*Carter et al., 2020*), and then the extracted lipids were esterified into fatty acid methyl esters (FAMEs) by heating at 70°C for 2 hr in 1M acetyl chloride in methanol. Duplicate 1 µL aliquots of sample FAMEs (I mg/mL in dichloromethane) were injected into a Shmadzu Scientific Instruments QP2010S GC-MS linked to a 2010 FID (Shmadzu Scientific Instruments, Kyoto, Japan) at Sacred Heart University (Fairfield, CT). Peaks were identified by retention times established by analysis of GLC standard FAME mixes (Nu-Chek Prep, Elysian, MN USA) run every 15 samples and visual inspection of all chromatograms. Concentrations of individual FAs were calculated as a percent by mass (FA peak area/total chromatogram area).

## Statistical analysis

All the analyses were performed in R (R Core Team 2018, R version 3.5.1) and prior to interpreting the output we checked model assumptions including inspection of residuals for homoscedasticity, normality and independence (lack of pattern). For Experiment I, we used general linear models to infer the fixed effect of Diet (MUFA vs. PUFA) on energy expenditure and changes in body mass and plasma metabolites during the 6 hr flight. Body mass was included as a fixed covariate to control for associated variation in energy expenditure. For Experiment II, we used linear mixed models (*Bates et al., 2015*) with restricted maximum likelihood (REML) estimation to infer the fixed effect of Diet (MUFA vs. PUFA), Flight training (trained vs. non-trained), and their interactive effect on BMR. Body mass and date of measurement were included as a fixed covariate and the number of measurement chamber as a random effect. Full models always included the interaction of main effects and covariates; however, whenever possible, we simplified the model by removing non-significant interactions. We also used linear mixed models with restricted maximum likelihood (REML) estimation to infer the fixed effect of Diet (MUFA vs. PUFA), Flight training (trained vs. non-trained) and time-point ('Pre-training' vs. 'Post-flight' vs. 'Recovery') on plasma metabolites and oxidative status, with individual bird included as a random effect. We used the Type III test with Kenward-Roger approximation method for the effective degrees of freedom, from the package 'lmerTest' (*Kuznetsova et al., 2017*). Descriptive statistics show adjusted least squares means with 95% confidence intervals (LSM ± 95% CI).

## Acknowledgements

We are thankful to the many members of the McWilliams lab who provided stimulating discussions on key topics. Major logistical support was provided by the animal care staff, technical staff, and scientists at the Max Planck Institute for Ornithology, Seewiesen, Germany. Special thanks to Carola Schmidt-Wellenburg and Ninon Ballerstaedt who helped with the windtunnel experiments, B Helm, B Kempenaers, H Gwinner, B Biebach, and M Starck for regular stimulating discussions, and H Biebach for having the foresight and fortitude to create the windtunnel at MPIO, Seewiesen. Funding: this research was supported by grants from the National Science Foundation to SRM and BJP (IBN-9983920 and IOS-1354187), the USDA to SRM (RIAES-538748), and an OPUS grant from the National Science Foundation (NCN), Poland, to UB (UMO-2015/19/B/NZ8/01394).

## Additional information

### Funding

| Funder | Grant reference number | Author |
| --- | --- | --- |
| National Science Foundation | IBN-9983920 | Scott McWilliams Barbara Pierce |
| U.S. Department of Agriculture | RIAES-538748 | Scott McWilliams |
| National Science Foundation | UMO-2015/19/B/NZ8/01394 | Ulf Bauchinger |
| National Science Foundation | IOS-1354187 | Scott McWilliams Barbara Pierce |

The funders had no role in study design, data collection and interpretation, or the decision to submit the work for publication.

### Author contributions

Scott McWilliams, Conceptualization, Resources, Data curation, Formal analysis, Supervision, Funding acquisition, Investigation, Methodology, Writing - original draft, Project administration, Writing - review and editing; Barbara Pierce, Conceptualization, Resources, Funding acquisition, Methodology, Project administration, Writing - review and editing; Andrea Wittenzellner, Resources, Investigation, Methodology, Project administration, Writing - review and editing; Lillie Langlois, Lisa Trost, Investigation, Methodology, Project administration, Writing - review and editing; Sophia Engel, Olivia Fatica, Investigation, Methodology, Writing - review and editing; John R Speakman, Wolfgang Goymann, Maciej Dzialo, Formal analysis, Investigation, Methodology, Writing - review and editing; Kristen DeMoranville, Data curation, Investigation, Methodology, Project administration, Writing - review and editing; Amadeusz Bryla, Edyta Sadowska, Data curation, Formal analysis, Investigation, Methodology, Writing - review and editing; Ulf Bauchinger, Conceptualization, Resources, Formal analysis, Supervision, Funding acquisition, Investigation, Methodology, Project administration, Writing - review and editing

### Author ORCIDs

Scott McWilliams https://orcid.org/0000-0002-9727-1151

John R Speakman http://orcid.org/0000-0002-2457-1823

Wolfgang Goymann http://orcid.org/0000-0002-7553-5910

Maciej Dzialo http://orcid.org/0000-0002-3632-8572

Edyta Sadowska http://orcid.org/0000-0003-1240-4814

### Ethics

Animal experimentation: All procedures adhered to the ethical guidelines of the North American Ornithological Council (Fair et al., 2010) and were approved by the University of Rhode Island IACUC (Protocols #AN09-09-009, AN08-02-014) and the Government of Upper Bavaria, Germany (AZ 55.2-1-54-2532-216-2014).

Decision letter and Author response
Decision letter https://doi.org/10.7554/eLife.60626.sa1
Author response https://doi.org/10.7554/eLife.60626.sa2

## Additional files

### Supplementary files
• Transparent reporting form

### Data availability
All data are available in the main text or the supplementary materials.

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
