## [Decision Letter]

**Acceptance summary:**

This study reports that migratory birds that consume omega-6 and -3 fats spend less energy during endurance flights; however, such short-term energy savings come at the cost of longer-term oxidative damage. This is the first demonstration of such an energy savings, oxidative cost trade-off in a migratory bird. An understanding of the ecological factors that mediate this trade-off has implications for the conservation physiology of migratory birds whose numbers are rapidly declining.

**Decision letter after peer review:**

Thank you for submitting your article "The energy savings-oxidative cost trade-off for birds during migration" for consideration by *eLife*. Your article has been reviewed by three peer reviewers, one of whom is a member of our Board of Reviewing Editors, and the evaluation has been overseen by Christian Rutz as the Senior Editor. The reviewers have opted to remain anonymous.

The reviewers have discussed their reviews with one another, and the Reviewing Editor has drafted this decision letter to help you prepare a revised submission.

Summary:

Your study used diet manipulations to raise two groups of European starlings Sturnus vulgaris with different fatty acid composition in their fat stores. These starlings were used to test two complementary hypotheses: (1) that birds with more essential omega-6 and -3 polyunsaturated fatty acids will have an enhanced flight performance; and (2) that enhanced flight performance leads to upregulation of the endogenous antioxidant system or incurs oxidative damage. Your results showed that birds composed of more essential omega-6 and -3 polyunsaturated fatty acids expended 11% less energy compared to birds with more monounsaturated fatty acids. These groups did not differ in Basal Metabolic Rate, however, birds with more essential omega-6 and -3 polyunsaturated fatty acids had higher longer-term oxidative damage. You, therefore, conclude that migrants face an "energy savings-oxidative cost" trade off.

All reviewers found the hypotheses interesting, the experiment well designed, the data analyses sound and the results compelling. However, although we do not think that additional data are required to validate your conclusions, we have major concerns about some of your claims and these will need to be carefully addressed during revision, especially with respect to migration and migratory birds.

Revisions:

1) The content of the study is more about endurance exercise rather than migration, and this makes the title restrictive and the conclusion somewhat unsubstantiated. This issue is even more apparent because nothing is stated about the species' migratory ecology. Please pay attention to the following specific comments:

a) It seems a bit daring to speak about "a trade-off which migrants face". You did not show whether the starlings were in a migratory state (e.g., increase in body mass and/or fat score, Zugunruhe). Therefore it is not clear, whether the starlings did prepare for migratory state and went through "up-regulation of many aspects of physiology", such as the up-regulation of oxidative enzymes (Lundgren and Kiessling, 1985, Oecologia 66). Since the starlings were not feeding their natural diet, it might be possible that they did not have the same high oxidative capacity as free-living migrants. It would be better to speak about "endurance flight" instead of migration. Another aspect which was not considered is that European starlings are short-distance migrants, or sedentary in southern Germany. It was shown in many studies that they differ in their migration strategies and physiological adaptions from long-distance migrants. Short-distance migrants probably catabolize more protein and/or carbohydrates than long-distance migrants. Hence, one might question whether short-distance migrants, such as the starlings, which have the opportunities for frequent stop-overs to recover and to refuel, do actually face this trade-off. Maybe they do not choose a PUFA-rich diet. These aspects should be discussed and relevant literature which is missing in the manuscript should be included.

b) The basis for using European starlings for testing the hypotheses was not stated nor justified in the study. In addition, there was no explanation for using 15 days of flight-training or 6 hours (260 km) of non-stop flight for the experiment. Is this equivalent to the standard migratory distance of the species or of the population from which the nestlings were collected for the experiment? The striking absence of this information obscures the ecological relevance of the study and makes it seem like a purely physiology curiosity-driven study, with little ecological application. We would like to see some information on the ecology of the species and why it is interesting/suitable for testing these hypotheses, which apply to animals beyond birds.

c) The title of the study is restricted to migration or migratory birds even though the study is about endurance exercise in animals. Nonetheless, it is still necessary to highlight the implication of the findings for migration.

d) Can you expand this paragraph a little to substantiate the statement "Collectively, our study provides compelling evidence that avian athletes – just like human athletes – face considerable trade-offs when deciding what to eat to enhance their performance", with human examples of diet optimisation or its implications for enhanced exercise performance versus oxidative damage? Similarly, can you expand the said paragraph by discussing the implication(s) of this trade-off for migratory and other high-performance bird species?

2) The methodological limitations of the study were not addressed in the Discussion. Please pay attention to, and address, these specific comments:

a) We found the main strength and novelty of the study to be the first experiment. The experiment is well designed and uses some complicated and impressive methodology (DLW, wind tunnel). DLW methodolgy has much improved over the last 20 years and become a gold standard for activity metabolic rate measurement. But, DLW is still based on some assumptions for RQ and water loss. Is it possible that 11% difference (on average) is related to a difference in RQ, water loss and lean mass? We are not sure, but it should be considered and discussed. In a perfect design, we would expect to see a comparison between BMR recorded by respirometry and DLW, which would eliminate this doubt.

b) The second experiment is less clear and refers to recent previous work by the group (Carter et al., 2020) where an effect of diet and time (training) on BMR was found. We suggest this should be discussed, as no such effect was found in the current study.

3) Please provide methodological details about the experimental design and the data analyses that will help readers understand Figure 3, which is the basis of your conclusions about the "energy savings-oxidative cost" trade-off. Please pay attention to the comments below:

a) You hypothesize that higher PUFAs in biological membranes lead to elevated risk of oxidative damage, and report higher oxidative damage for the high PUFA group. Here, we had some difficulties to understand the results. We found Figure 3C to be confusing - in which of the three time points in the experiment is there a difference in oxidative stress? We would expect that the most significant effect of diet should be right after intensive aerobic activity, but from the data, we could not understand when the oxidative damage was recorded or if it was a summary of the three time points. Maybe we missed something, but this requires clarification, as this is the second most important finding of this work that should support the proposed trade-off hypothesis.

b) The Materials and methods and Results are not clearly presented. The terms are highly confusing, the legends do not explain all terms. For example, it is unclear how the mean values of the MUFA and PUFA groups in Figure 3 were calculated.

c) The experimental design should be described in more detail. Please give the physiological state of the birds (fasted, post-resorptive) when sampled, and the time elapsed after flight until the end of blood sampling; both factors crucially affect plasma metabolites. Please explain what happened to the starlings after the experiment and how you analysed the fatty acids (tissue, method, etc).

d) Results and Discussion paragraph six: The terms are confusing. The "trained" group includes the untrained control birds. Therefore, it would be better to call the untrained birds "controls" or "sedentary" throughout the manuscript. The "post-flight" and "trained" birds were both sampled after the 6 hours flight. Since "trained" birds were sampled 2 days "post-flight", the term "recovery" might be clearer. Another possibility would be to call them Flyers/Controls day 0, day 16, and day 18.

Results and Discussion paragraph seven, Figure 3A: According to Figure 3A, uric acid of the flyers was different between the two groups, for the flyers and the controls. Please correct.

Figure 3B: It is strange that the controls showed a decrease of antioxidant capacity within the 2 days post-flight. Please explain and discuss this result.

Figure 3, right panel: Did you take the average per diet over all 3 groups? Please explain this here and/or in the Materials and methods.

Figure 3: Please explain LSM in the legend.

Right panels with the main effect of diet on oxidative status: To which group do these values refer, pre-training, post-flight, or trained?

Figure 3: Sedentary birds: How do you explain the change of uric acid and AO in sedentary birds over time? They were just sitting in their cages. AO and uric acid in sedentary birds should remain at the same level, assuming they were in the same physiological state. Uric acid reflects diet composition and physiological state. Since the birds received the same diet throughout the experiment and did not do flight training, this is hard to explain.

4) Please revise the Discussion to focus on the core finding of the experiment, which is the occurrence of an "energy savings - oxidative cost" trade-off during endurance flight due to fatty acid composition of body fat stores. You can highlight the implications of this finding for understanding migratory birds and other animals that are likely to undertake such endurance flights. Please pay attention to the comments below:

a) Since birds may choose among food items to optimize the trade-off between enhanced flight performance and long-term oxidative damage in natural conditions, it may be important to highlight the ecological conditions under which this "energy savings - oxidative cost" trade-off will be detrimental. This would be interesting to a readership interested in subjects like the ecological factors (e.g, foraging) responsible for the survival of many declining long-distance migrants during migration.

b) Relevant literature is missing. There are other studies that measured oxidative stress in birds during endurance flight and on migration (e.g., Costantini et al., 2008; Jenni-Eiermann et al., 2014).

c) Relevant aspects regarding migrating birds are missing in the Discussion:

Introduction: Above you mention that birds during endurance flight derive their energy mainly from fats and that these fats are highly "susceptible to oxidative damage". Since other non-avian athletes derive their energy mainly from carbohydrates (e.g., humans) and varying amounts of protein, the effect of the catabolism of these other fuels on oxidative stress should also be considered. Literature data showed a relationship between anti-oxidants and ROS with muscle score for free-living European robins during migratory flight, indicating the importance of muscle degradation (Jenni-Eiermann et al., 2014).

"Differences in the energy costs…". If known, please give the cause why the cyclists differ in their energy costs. Otherwise this sentence can be omitted.

d) Flying birds: Why should uric acid in trained birds be lower than pre-training? During flight, birds catabolize proteins and uric acid increases; during the recovery days, the UA levels should return to basal values. This is very strange. Please discuss this result. It is also hard to understand why the oxidant capacity does not recover, especially because the birds did not experience oxidative damage.

e) The composition of the diet may change between resting places and it also differs between spring and autumn migration. Maybe the migrants "return" to a less PUFA-rich diet after migratory bouts. In that case, the oxidative damage would be reduced. Please consider and discuss this aspect. There is a vast literature on this.

---

## [Author Response]

Revisions:1) The content of the study is more about endurance exercise rather than migration, and this makes the title restrictive and the conclusion somewhat unsubstantiated. This issue is even more apparent because nothing is stated about the species' migratory ecology. Please pay attention to the following specific comments:a) It seems a bit daring to speak about "a trade-off which migrants face". You did not show whether the starlings were in a migratory state (e.g., increase in body mass and/or fat score, Zugunruhe). Therefore it is not clear, whether the starlings did prepare for migratory state and went through "up-regulation of many aspects of physiology", such as the up-regulation of oxidative enzymes (Lundgren and Kiessling, 1985, Oecologia 66). Since the starlings were not feeding their natural diet, it might be possible that they did not have the same high oxidative capacity as free-living migrants. It would be better to speak about "endurance flight" instead of migration. Another aspect which was not considered is that European starlings are short-distance migrants, or sedentary in southern Germany. It was shown in many studies that they differ in their migration strategies and physiological adaptions from long-distance migrants. Short-distance migrants probably catabolize more protein and/or carbohydrates than long-distance migrants. Hence, one might question whether short-distance migrants, such as the starlings, which have the opportunities for frequent stop-overs to recover and to refuel, do actually face this trade-off. Maybe they do not choose a PUFA-rich diet. These aspects should be discussed and relevant literature which is missing in the manuscript should be included.

Thank you for carefully considering the most appropriate context for the results we report. As requested, we have added information about the migratory disposition of the population of starlings from which the birds we used were sampled, and we have made some substantial changes to the text as outlined below. We have also modified the title, as requested, to be less restrictive.

Regarding the specific comments of the reviewers, we acknowledge that any controlled experiment such as ours by definition controls for many potentially confounding variables at the potential expense of some extrapolation to natural conditions. Here we make the case that we went to great lengths to make our experiment as relevant to migratory birds as possible, although the reviewers comments also helped to improve the manuscript along these lines. Specifically, as detailed below and in the revised manuscript, the design of our experiment involved hand-raising nestlings taken from a population of free-living, migratory birds. Furthermore, we used prepared diets that had the same macronutrient composition as natural fruits regularly eaten by wild birds – such prepared diets ensured that the diet composition was entirely known and consistent, that it was ecologically relevant and, most importantly for our study, that the two experimental diets differed only in their fatty acid composition. As we have shown, this difference in fatty acid composition of diets produced starlings with fat stores that were similar in fatty acid composition to that observed in free-living songbirds during migration. In addition, we trained birds to fly in one of three windtunnels in the world designed for flying songbirds for long durations so that we could accurately measure energy expenditure during flight and the associated oxidative costs. We agree that some readers may question the extent to which our results from this controlled experiment apply to free-living migratory birds; however, we emphasize in the revised manuscript these many aspects of the experiment that were designed to increase the ecological relevance of the work. Thank you to the reviewers for raising these important points and for the opportunity to revise the original work along these lines.

Below we outlined in detail how we revised the article to address these important points.

We knew from earlier banding studies that the starling population in southern Germany from which our nestlings were taken was migratory. Thus, we have added the following to the Materials and methods section to emphasize that this population of starlings is migratory and so appropriate for our study:

“European starlings from this southeastern German population are medium-distance diurnal migrants that leave for wintering grounds in October and November and return to their Bavarian breeding grounds in April (48). Migratory distances for this population vary from many hundreds of kilometers to several thousand kilometers, with some individuals overwintering in the Euro-Mediterranean region and others in northwest Africa (48). European starlings are also quite social and curious, and quickly learn to successfully fly together in a given windtunnel as demonstrated by several recent studies (34, 49).”

We also carefully formulated the two experimental diets to have the macronutrient composition of high-lipid fruits that are often eaten by fall-migrating songbirds including European starlings. We added the following text related to the point:

“The macronutrient composition of the two semi-synthetic diets simulates a natural high-lipid fruit diet (53, 54), and the primary fatty acids in the diet (>90% 16:0, 18:1, and 18:2) are also the most common fatty acids in natural fruits (21) and in songbirds that eat fruits during migration (21, 45). European starlings are like many fall-migrating songbirds in that they switch to eating largely fruits (46), much to the chagrin of many vineyard owners.”

We also knew from previous studies, including some of our own, that the fatty acid composition of fat stores in starlings that we produced using the two experimental diets was similar to that of freeliving migratory songbirds. Thus, we added the following to the main text and the Materials and methods section to emphasize the ecological relevance of this fatty acid composition of starlings used in our study:

“Such differences in fatty acid composition of fat stores in starlings, specifically the relative amounts of 18:1, 18:2, and 18:3, are also among the primary longer-chain fatty acids that compose the fat stores of wild songbirds especially during migration (18, 21, 22).”

“Previous work has established that these are the primary longer-chain fatty acids in wild songbirds especially during migration (18, 21, 22), and that the fatty acid composition of birds reflects that of their diet (13, 18, 19) which we also demonstrate here (Table 2).”

We certainly agree with the reviewer(s) that the migration strategy of birds (i.e., whether short- or long-distance migrants) can influence important aspects of how often individuals must rest and refuel, and how far they may travel in a given day/night. However, we respectfully disagree that birds with different migration strategies may mobilize different amounts of proteins, carbohydrates, and lipids during a given long-duration flight. Guglielmo, 2018, recently reviewed this topic and describes how glycogen and amino acid stores can power flight for only very short periods – approximately 5 minutes for glycogen and slightly longer for amino acids. Furthermore, migratory birds have been shown to upregulate fatty acid transport proteins to promote fatty acid uptake within muscle cells where fat is also stored – these stored lipids can thus be quickly mobilized to fuel flight. Therefore, birds regardless of migration strategy use primarily lipids to fuel flights longer than ~30 minutes, something that makes them exceptional endurance athletes. Furthermore, others have compelling argued that distinguishing between short- and long-distance migrants does not result in clear differences when considering most behavioural and physiological traits, rather migratory traits appear as a continuum with no clear distinction necessary (Piersma et al. 2005 pp. 282-293 *in* Bauchinger et al., editors. Bird Hormones and Bird Migrations: Analyzing Hormones in Droppings and Egg Yolks and Assessing Adaptations in Long-Distance Migration.)

We emphasize this point in the text:

“birds use fats as their primary fuel (about 95%) for high-intensity endurance exercise such as migratory flights (8-10)”

We have also carefully considered each instance in the article where we use the terms “migratory flight(s)” to determine when it is more appropriate and accurate to use “endurance flight(s)”. We thank the reviewer for drawing our attention to this inappropriate usage. For example, we have modified the title so it now reads:

“The energy savings-oxidative cost tradeoff for migratory birds during endurance flight.” as opposed to the original “The energy savings-oxidative cost tradeoff for birds during migration.”

Finally, for logistical reasons we did not directly measure food intake of starlings over the course of the fall experiments. The use of *Zugunruhe* (nocturnal activity) as an indicator of “migration state” is not appropriate as starlings are diurnal migrants. We have emphasized this point in the text:

“We did not directly verify that such decreases in light levels in fall induced starlings to increase food intake, although many other studies provide such evidence in migratory birds (55-57), or to increase Zugunruhe (nocturnal activity), since starlings are diurnal migrants.”

b) The basis for using European starlings for testing the hypotheses was not stated nor justified in the study. In addition, there was no explanation for using 15 days of flight-training or 6 hours (260 km) of non-stop flight for the experiment. Is this equivalent to the standard migratory distance of the species or of the population from which the nestlings were collected for the experiment? The striking absence of this information obscures the ecological relevance of the study and makes it seem like a purely physiology curiosity-driven study, with little ecological application. We would like to see some information on the ecology of the species and why it is interesting/suitable for testing these hypotheses, which apply to animals beyond birds.

As noted above, we acknowledge that any controlled experiment such as ours by definition controls for many potentially confounding variables at the potential expense of some extrapolation to natural conditions. Here we make the case that we went to great lengths to make our experimental results as relevant to migratory birds as possible, and we have revised the article (specified above) to include text that better makes this case so that readers can better judge the extent to which our results are ecologically relevant. In short, the design of our experiment involved hand-raising nestlings taken from a population of free-living, migratory birds, and we fed starlings prepared diets that simulated the macronutrient composition of natural fruits regularly eaten by wild birds including starlings. Furthermore, we trained birds to fly in one of three windtunnels in the world designed for flying songbirds for long durations so that we could accurately measure energy expenditure during flight and oxidative costs. We trust that this additional information in the revised manuscript on the ecology of these European Starlings will convince readers of the ecological applications of this work.

Regarding the explanation for the training schedule and long-duration flight, we have added the following text:

“The flight-training schedule and the duration of the longest (6 hr) flight were chosen based on logistics as well as ecological relevance. The 15-day flight-training schedule and flying conditions (always 12 m s^-1^ wind speed and 15 °C) had been used in previously successful experiments at MPIO designed to fly barn swallows (Hirundo rustica) and starlings (Sturnus vulgaris, Sturnus roseus) for long durations (47, 50, 51, 58, 60), and free-living songbirds including starlings typically complete their migration from breeding to wintering areas over many days of flying and stopovers (46, 61). The duration of the final longest flight (6-hrs, 260 km) was sufficiently long to provide adequate turnover of the isotope-labelled water and so ensure accurate measurements of energy expenditure using the doubly-labelled water (DLW) technique (62). The 6-hr duration of flight is also within the range of typical single-day migratory flights for many free-living songbirds including starlings (46, 61).”

In sum, we thank the reviewers for requesting this additional information because it does make clearer the ecological relevance of the work.

c) The title of the study is restricted to migration or migratory birds even though the study is about endurance exercise in animals. Nonetheless, it is still necessary to highlight the implication of the findings for migration.

Thank you for this suggestion. We have modified the title so it now reads “The energy savings-oxidative cost tradeoff for migratory birds during endurance flight.”

d) Can you expand this paragraph a little to substantiate the statement "Collectively, our study provides compelling evidence that avian athletes – just like human athletes – face considerable trade-offs when deciding what to eat to enhance their performance", with human examples of diet optimisation or its implications for enhanced exercise performance versus oxidative damage? Similarly, can you expand the said paragraph by discussing the implication(s) of this trade-off for migratory and other high-performance bird species?

Thank you for the opportunity to extend our Discussion of the implications of our work. There is a large literature on the importance of nutrition in determining human exercise performance, and especially in the last 10 yrs quite a bit of attention on dietary fatty acid composition and antioxidants on human exercise performance. We have added several references to this sentence that substantiate this statement and point the readers to key literature on this topic. Also as requested, we have expanded this paragraph to include other implications of this trade-off for migratory birds. Specifically, this section of the paragraph now reads as followed:

“ …. birds may choose among diets to optimize the trade-off between enhanced flight performance (more 18:2) while reducing the long-term costs of being composed of more longchain PUFA. Migratory birds can also optimize this energy savings-oxidative cost tradeoff by being composed of more n-3 and/or n-6 PUFAs only during migration periods when energy demands and fat catabolism are most extreme, and then become more monunsatured in composition during non-migration periods – such seasonal changes in fatty acid composition are commonly observed in migratory birds (21, 45).”

2) The methodological limitations of the study were not addressed in the Discussion. Please pay attention to, and address, these specific comments:a) We found the main strength and novelty of the study to be the first experiment. The experiment is well designed and uses some complicated and impressive methodology (DLW, wind tunnel). DLW methodolgy has much improved over the last 20 years and become a gold standard for activity metabolic rate measurement. But, DLW is still based on some assumptions for RQ and water loss. Is it possible that 11% difference (on average) is related to a difference in RQ, water loss and lean mass? We are not sure, but it should be considered and discussed. In a perfect design, we would expect to see a comparison between BMR recorded by respirometry and DLW, which would eliminate this doubt.

Thank you for this inciteful comment. As pointed out by the reviewer(s), differences in rates of water loss during flights can significantly affect the DLW estimates of energy expenditure. This was unlikely in our study given that the relative humidity and temperature and duration of the 6-hr flights in the windtunnel were tightly controlled. We have stated the following to make sure such a concern is alleviated:

“A single-pool model was used to calculate rates of CO_2_ production as recommended for use in animals less than 5 kg in body mass (69). There are several approaches for the treatment of evaporative water loss in the calculation (70). We detected no differences in body water pool before and after 6-hr flights. We assumed evaporation of 25% of the water flux (equation 7.17 in Speakman (62), which minimizes error in a range of conditions (70, 71).”

Also as pointed out by the reviewer(s), a key assumption of the DLW that is most relevant to our study is that the substrates used as fuel (carbohydrate, fat, protein) are known – this is important because the DLW technique estimates carbon dioxide production (not oxygen consumption) and carbon dioxide production depends on fuel type (unlike oxygen consumption). Given that our experimental design involved feeding birds diets that were identical in macronutrient composition and differed only in their fatty acid composition, the dietary substrates (carbohydrates, fats, proteins) available as fuels were also identical. Although it was not possible to directly measure RQ or change in lean and fat mass of individuals during the 6-hr flights, the effect of variation in the amount of stored fat vs. protein used as fuel (as reflected in the RQ) results in errors in estimates of energy expenditure not exceeding ± 2% (Black et al., 1986, Schmidt-Wellenburg et al., 2008, Westerterp, 2017). We have added the following to the description of the DLW method to emphasize this point:

“Given that our experimental design involved feeding birds diets that were identical in macronutrient composition and differed only in their fatty acid composition, the dietary substrates (carbohydrates, fats, proteins) available as fuels were also identical. Empirically measured variation in the amount of stored fat or protein used as fuel (as reflected in the RQ) results in errors in estimates of energy expenditure not exceeding ± 2% (50, 72, 73).”

b) The second experiment is less clear and refers to recent previous work by the group (Carter et al., 2020) where an effect of diet and time (training) on BMR was found. We suggest this should be discussed, as no such effect was found in the current study.

We trust that the substantial clarifications we have made to the description of the experimental design and methods (outlined throughout this letter), plus the revised main text describing the results will make the results from Experiment II much clearer. Once again, thank you for the opportunity to make these clarifying changes.

The reviewer is correct that we recently published results from another experiment conducted in Canada by our group (i.e., Carter et al., 2020) – we had referenced this study (#34) in the original draft of this manuscript. In that study, we fed Canada-born European Starlings one of two diets that differed primarily in 18:2n-6 and this in turn produced starlings with corresponding differences in fatty acid composition of fat stores and muscle membranes that were consistent over the 4-month fall experiment. Furthermore, birds with higher concentrations of 18:2n-6 in membranes and fat stores had higher BMR and peak metabolic rates, although this pattern was evident only early in the fall and not later in the fall experiment. This study did not include starlings that flew under controlled conditions in a wind-tunnel for 6-hrs at a time, unlike the present manuscript that we have submitted for your consideration. The change through time in these performance measures but not membrane composition led Carter et al., 2020, to conclude that their results were most consistent with the signal hypothesis compared to other competing hypotheses – we had also discussed this result in the original draft of this manuscript. Given that we had already discussed these results in the original version of the manuscript, we have not added any further discussion of this other published work in the revised draft of this manuscript.

3) Please provide methodological details about the experimental design and the data analyses that will help readers understand Figure 3, which is the basis of your conclusions about the "energy savings - oxidative cost" trade-off. Please pay attention to the comments below:a) You hypothesize that higher PUFAs in biological membranes lead to elevated risk of oxidative damage, and report higher oxidative damage for the high PUFA group. Here, we had some difficulties to understand the results. We found Figure 3C to be confusing, in which of the three time points in the experiment is there a difference in oxidative stress? We would expect that the most significant effect of diet should be right after intensive aerobic activity, but from the data, we could not understand when the oxidative damage was recorded or if it was a summary of the three time points. Maybe we missed something, but this requires clarification, as this is the second most important finding of this work that should support the proposed trade-off hypothesis.

Thank you for making sure that this result is clear to readers, as we agree this is an important finding of our work.

The experimental design included two diet groups (MUFA and PUFA) and two exercise groups (flight-trained and untrained, “sedentary” control individuals). Birds in each of these four groups were blood sampled at three times over the course of the flight training period (prior to any training, immediately after a long-duration flight, and two days later after the birds had recovered from any acute effect of flight). Importantly, sedentary control birds were blood sampled on the same days as the flight-trained birds in their cohort to control for any coincident time-of-fall effects.

To make this clearer, we have revised the legend to Table 3 (where the main statistical results are reported) so that it now reads:

“Table 3. The effect of flight (flight-trained for 15 days in wind-tunnel or not; Trained or Sedentary), diet (MUFA or PUFA), and time (blood sampled at three different time points: before the start of flight training in the wind-tunnel (“pre-training”), immediately after a long duration flight on Day 15 (“post-flight”), and ca. 1.5 days afterwards (“recovery”)) on plasma metabolites and oxidative status in European starlings in Experiment II. Note that individuals that did not undergo flight training (i.e., control “untrained” birds) were sampled on the same days as flight-trained birds in their same cohort. Test statistics: F-value with denominator degrees of freedom (ddf) and significance level p-value for main factors and their interactions from the linear mixed models.”

We have also revised Figure 3 to make clearer the comparisons and key results. We corrected a small typo in the reporting of the multiple comparisons test in Figure 3B which now makes clear that the sedentary control birds did not show a decrease in antioxidant capacity over the flight training period. We also added a new header to the right panel in Figure 3 (“Main effect of diet”) that should make crystal clear that these panels are presenting the key diet main effects.

As outlined elsewhere in this letter, we have also made substantial changes to the text of the Materials and methods section to make clearer the experimental design and methods.

b) The Materials and methods and Results are not clearly presented. The terms are highly confusing, the legends do not explain all terms. For example, it is unclear how the mean values of the MUFA and PUFA groups in Figure 3 were calculated.

We of course did not intend for the Materials and methods and Results to be confusing. In order to more clearly present the results, we have adopted the suggestions of the reviewer for the renaming of the groups. In short, we revised the manuscript terms throughout to now refer to “trained” or “flight-trained” birds versus “sedentary” control birds (instead of the original “untrained” term). Likewise, the three time points are now referred throughout as “Pre-training”, “Post-flight”, and “Recovery” (instead of the original “trained” term for the latter). Thank you for these suggestions.

We have also substantially revised the legends to Figure 3 and Table 3 to make clearer the comparisons and key results. In short, the experimental design included two diet groups (MUFA and PUFA) and two exercise groups (flight-trained and untrained, “sedentary” control individuals). Birds in each of these four groups were blood sampled at three times over the course of the flight training period (prior to any training, immediately after a long-duration flight, and two days later after the birds had recovered from any acute effect of flight). Importantly, sedentary control birds were blood sampled on the same days as the flight-trained birds in their cohort to control for any coincident time-of-fall effects. The key results are shown in Figure 3 and the details of the statistical analyses are reported in Table 3. This right panel in Figure 3 reports the main effect of diet (MUFA vs. PUFA) on the three measures. To make this crystal clear, we have added a new header in Figure 3 associated with this right panel, “Main effect of diet”.

The legend to Table 3 has also been revised and now reads:

“Table 3. The effect of flight (flight-trained for 15 days in wind-tunnel or not; Trained or Sedentary), diet (MUFA or PUFA), and time (blood sampled at three different time points: before the start of flight training in the wind-tunnel (“pre-training”), immediately after a long duration flight on Day 15 (“post-flight”), and ca. 1.5 days afterwards (“recovery”)) on plasma metabolites and oxidative status in European starlings in Experiment II. Note that individuals that did not undergo flight training (i.e., control “untrained” birds) were sampled on the same days as flight-trained birds in their same cohort. Test statistics: F-value with denominator degrees of freedom (ddf) and significance level p-value for main factors and their interactions from the linear mixed models.”

c) The experimental design should be described in more detail. Please give the physiological state of the birds (fasted, post-resorptive) when sampled, and the time elapsed after flight until the end of blood sampling; both factors crucially affect plasma metabolites. Please explain what happened to the starlings after the experiment and how you analysed the fatty acids (tissue, method, etc).

Thank you for the opportunity to provide more details about the experimental methods.

First, all blood samples used for metabolites were taken after birds had been fasted for at least 12 hrs (i.e., after an overnight without food and before offered food on the Pre-training and Recovery days, and after an overnight without food plus their longest flight on the Post-flight day). We have included this information in the Materials and methods in section “7. Blood sampling and analyses”.

Second, all blood samples used for metabolites were taken within 3 min of capture, including within 3 min of completing the longest duration flight. We have also included this information in the Materials and methods in section “7. Blood sampling and analyses”.

Lastly, apologies for this omission about how the fat stores were sampled from starlings, and about the fatty acid analyses. We have added a section “8. Fatty acid composition of diet and furcular fat” to the Materials and methods in which we describe these methods, as follows:

“Fatty acid composition of diet and furcular fat

Fatty acid composition of the two semi-synthetic diets (Table 2) was measured by gas chromatography in lipids extracted using a modified Folch method (34) from composited subsamples of each diet taken over the course of the experiments. We collected adipose tissue from the visible fat stores in the furcular area of starlings used in Experiment I (n = 29) 5-7 days after their longest flight in the wind-tunnel. We conducted the biopsies of this furcular fat using the methods outlined in Rocha et al. (82). Briefly, upper breast feathers were wetted and moved aside to expose the skin and visible subcutaneous yellow-colored fat stores. We selected an area without visible capillaries, disinfected the area with antiseptic solution, and applied a topical anaesthetic gel to this area. After ca. 10 min, we then pinched the skin in this area to ensure no pain response by the bird. We then made an ca. 3 mm-long incision in the skin using a sharp scalpel, pulled a small piece of adipose tissue (ca. 10-20 mg) through the incision using sterile forceps, and cut the sample under the forceps using sterile scissors. The incision area was then realigned and a thin layer of veterinary tissue glue was applied to seal the incision. All birds were checked weekly thereafter and the wound was completely healed within 2-3 weeks.

Dietary fat and furcular fat samples were stored at -80 C until fatty acid composition was measured by gas chromatography. We thawed samples, extracted total lipids using a modified Folch method (34), and then the extracted lipids were esterified into fatty acid methyl esters (FAMEs) by heating at 70 C for 2 hr in 1M acetyl chloride in methanol. Duplicate 1 ul aliquots of sample FAMEs (I mg/ml in dichloromethane) were injected into a Shmadzu Scientific Instruments QP2010S GC-MS linked to a 2010 FID (Shmadzu Scientific Instruments, Kyoto, Japan) at Sacred Heart University (Fairfield, CT). Peaks were identified by retention times established by analysis of GLC standard FAME mixes (Nu-Chek Prep, Elysian, MN USA) run every 15 samples and visual inspection of all chromatograms. Concentrations of individual FAs were calculated as a percent by mass (FA peak area/total chromatogram area).”

d) Results and Discussion paragraph six: The terms are confusing. The "trained" group includes the untrained control birds. Therefore, it would be better to call the untrained birds "controls" or "sedentary" throughout the manuscript. The "post-flight" and "trained" birds were both sampled after the 6 hours flight. Since "trained" birds were sampled 2 days "post-flight", the term "recovery" might be clearer. Another possibility would be to call them Flyers/Controls day 0, day 16, and day 18.

Apologies for the confusion of terms. Of course for us these short-hand terms made total sense as we conducted the experiment, although we now see how the naming of these experimental groups could be improved. We have adopted the suggestions of the reviewer for the renaming of the groups. In short, we revised the manuscript terms throughout to now refer to “trained” or “flight-trained” birds versus “sedentary” birds (instead of the original “untrained” term). Likewise, the three time points are now referred throughout as “Pre-training”, “Post-flight”, and “Recovery” (instead of the original “trained” term for the latter). Thank you for these suggestions.

Results and Discussion paragraph seven, Figure 3A: According to Figure 3A, uric acid of the flyers was different between the two groups, for the flyers and the controls. Please correct.Figure 3B: It is strange that the controls showed a decrease of antioxidant capacity within the 2 days post-flight. Please explain and discuss this result.

Thank you for catching these two errors – we have corrected what we wrote about the uric acid results, and we realized there was a small typo in the reporting of the multiple comparisons test in Figure 3B which now makes clear that the sedentary control birds did not show a decrease in antioxidant capacity over the flight training period. The sentence now reads:

“Flight-training over more than two weeks did not affect baseline levels of oxidative damage (compare Pre-training and Recovery; Figure 3C) while antioxidant capacity decreased in flight-trained but not sedentary birds (Figure 3B) and plasma uric acid decreased over time in both trained and sedentary starlings (Figure 3A; see Table 3 for detailed statistical results).”

Figure 3, right panel: Did you take the average per diet over all 3 groups? Please explain this here and/or in the Materials and methods.

This right panel in Figure 3 reports the main effect of diet (MUFA vs. PUFA) on the three measures. The detailed statistical results are shown in Table 3. We have cited this table and revised the legend to this figure to make this clearer. We have also included a new header in Figure 3 associated with this right panel, “Main effect of diet”.

Figure 3: Please explain LSM in the legend.Right panels with the main effect of diet on oxidative status: To which group do these values refer, pre-training, post-flight, or trained?

This right panel in Figure 3 reports the main effect of diet (MUFA vs. PUFA) on the three measures. The detailed statistical results are shown in Table 3. We have cited this table and revised the legend to this figure to make this clearer. We have also included a new header in Figure 3 associated with this right panel, “Main effect of diet”. Thank you for pointing out that we needed to define/explain the “LSM” acronym in the legend – we have done so.

Figure 3: Sedentary birds: How do you explain the change of uric acid and AO in sedentary birds over time? They were just sitting in their cages. AO and uric acid in sedentary birds should remain at the same level, assuming they were in the same physiological state. Uric acid reflects diet composition and physiological state. Since the birds received the same diet throughout the experiment and did not do flight training, this is hard to explain.

Thank you for carefully considering these results – as noted above, we have corrected what we wrote about the uric acid results, and we realized there was a small typo in the reporting of the multiple comparisons test in Figure 3B which now makes clear the sedentary control birds did not show a decrease in antioxidant capacity over the flight training period. The sentence now reads:

“Flight-training over more than two weeks did not affect baseline levels of oxidative damage (compare Pre-training and Recovery; Figure 3C) while antioxidant capacity decreased in flight-trained but not sedentary birds (Figure 3B) and plasma uric acid decreased over time in both trained and sedentary starlings (Figure 3A; see Table 3 for detailed statistical results).”

Like the reviewer, we did not expect the decrease in plasma uric acid over time in both sedentary and flight-trained birds, although now this remains the only one of the plasma metabolites that shows this trend. We have no good explanation for why plasma uric acid decreased in sedentary (and flight-trained) birds over time, although such evidence makes clear the importance of including this sedentary control group.

4) Please revise the Discussion to focus on the core finding of the experiment, which is the occurrence of an "energy savings - oxidative cost" trade-off during endurance flight due to fatty acid composition of body fat stores. You can highlight the implications of this finding for understanding migratory birds and other animals that are likely to undertake such endurance flights. Please pay attention to the comments below:a) Since birds may choose among food items to optimize the trade-off between enhanced flight performance and long-term oxidative damage in natural conditions, it may be important to highlight the ecological conditions under which this "energy savings - oxidative cost" trade-off will be detrimental. This would be interesting to a readership interested in subjects like the ecological factors (e.g, foraging) responsible for the survival of many declining long-distance migrants during migration.

Thank you for this suggestion. We have further discussed the ecological conditions under which this tradeoff can be detrimental as follows:

“…birds may choose among diets to optimize the trade-off between enhanced flight performance (more 18:2) while reducing the long-term costs of being composed of more longchain PUFA. Migratory birds can also optimize this energy savings-oxidative cost tradeoff by being composed of more n-3 and/or n-6 PUFAs only during migration periods when energy demands and fat catabolism are most extreme, and then become more monounsaturated in composition during non-migration periods – such seasonal changes in fatty acid composition are commonly observed in migratory birds (21, 45). Such a tradeoff may become especially detrimental if foods with different quantities of micronutrients (notably long-chain PUFAs and antioxidants) are not available in nature across the seasons, in which case birds may be unable to ameliorate such a tradeoff through careful choices of diet.”

b) Relevant literature is missing. There are other studies that measured oxidative stress in birds during endurance flight and on migration (e.g., Costantini et al., 2008; Jenni-Eiermann et al., 2014).

Thank you for allowing us the opportunity to emphasize that there have been a few other studies that have measured the oxidative challenges of endurance exercise such as migratory flights. We were quite aware of the original contributions provided by these two papers (Costantini et al., 2008; Jenni-Eiermann et al., 2014) and we were sure to cite their contributions in the revised manuscript – see, for example:

“Increased energy metabolism during exercise is often associated with increased production of pro-oxidants regardless of the fuel types used (i.e., carbohydrates, protein, fats) which causes oxidative damage if not quickly quenched by dietary antioxidants and/or by increased production of antioxidant enzymes (e.g., superoxide dismutase, glutathione peroxidase).”

c) Relevant aspects regarding migrating birds are missing in the Discussion:Introduction: Above you mention that birds during endurance flight derive their energy mainly from fats and that these fats are highly "susceptible to oxidative damage". Since other non-avian athletes derive their energy mainly from carbohydrates (e.g., humans) and varying amounts of protein, the effect of the catabolism of these other fuels on oxidative stress should also be considered. Literature data showed a relationship between anti-oxidants and ROS with muscle score for free-living European robins during migratory flight, indicating the importance of muscle degradation (Jenni-Eiermann et al., 2014).

Thank you for allowing us the opportunity to emphasize that the oxidative challenges of endurance exercise such as migratory flights of birds are evident independent of fuel type. We were quite aware of the original contributions provided by these two papers (Costantini et al., 2008; Jenni-Eiermann et al., 2014, and we were sure to cite their contributions in the revised manuscript:

“Increased energy metabolism during exercise is often associated with increased production of pro-oxidants regardless of the fuel types used (i.e., carbohydrates, protein, fats) which causes oxidative damage if not quickly quenched by dietary antioxidants and/or by increased production of antioxidant enzymes (e.g., superoxide dismutase, glutathione peroxidase) (5, 36-41).”

"Differences in the energy costs…". If known, please give the cause why the cyclists differ in their energy costs. Otherwise this sentence can be omitted.

When we have presented this work to audiences at meetings there is often a question about the relevance of an 11% energy savings. Thus, we would prefer to retain this one sentence that points out that the top-placing Tour de France cyclists differ by <5% in energy costs in hopes of emphasizing to readers the relevance of an 11% energy savings for birds composed of more PUFA.

d) Flying birds: Why should uric acid in trained birds be lower than pre-training? During flight, birds catabolize proteins and uric acid increases; during the recovery days, the UA levels should return to basal values. This is very strange. Please discuss this result. It is also hard to understand why the oxidant capacity does not recover, especially because the birds did not experience oxidative damage.

Thank you for carefully considering these results – as noted above, we realized there was a small typo in the reporting of the multiple comparisons test in Figure 3b which now makes clear the sedentary control birds did not show a decrease in antioxidant capacity over the flight training period. The sentence now reads:

“Flight-training over more than two weeks did not affect baseline levels of oxidative damage (compare Pre-training and Recovery; Figure 3C) while antioxidant capacity decreased in flight-trained but not sedentary birds (Figure 3B) and plasma uric acid decreased over time in both trained and sedentary starlings (Figure 3A; see Table 3 for detailed statistical results).”

Thus, there was no significant change in oxidative capacity in sedentary, control birds over time.

Like the reviewer, we did not expect the decrease in plasma uric acid over time in both sedentary and flight-trained birds. We have no good explanation for why plasma uric acid decreased in sedentary (and flight-trained) birds between the Pre-Flight and Recovery time points. However, we note that Table 3, which is referenced in the text, reveals that this is the only one of the three measures shown in Figure 3 with a significant Flight X Time interaction term. It is the remarkable increase in plasma uric acid during a given long-duration flight (in only flight-trained birds) that we have chosen to emphasize.

e) The composition of the diet may change between resting places and it also differs between spring and autumn migration. Maybe the migrants "return" to a less PUFA-rich diet after migratory bouts. In that case, the oxidative damage would be reduced. Please consider and discuss this aspect. There is a vast literature on this.

Thank you for pointing this out. We whole-heartedly agree that availability of PUFA-rich diets may differ over space and time, and especially between spring and autumn migration. We have briefly discussed this important point as follows:

“...birds may choose among diets to optimize the trade-off between enhanced flight performance (more 18:2) while reducing the long-term costs of being composed of more longchain PUFA. Migratory birds can also optimize this energy savings-oxidative cost tradeoff by being composed of more n-3 and/or n-6 PUFAs only during migration periods when energy demands and fat catabolism are most extreme, and then become more monounsaturated in composition during non-migration periods – such seasonal changes in fatty acid composition are commonly observed in migratory birds (21, 45). Such a tradeoff may become especially detrimental if foods with different quantities of micronutrients (notably long-chain PUFAs and antioxidants) are not available in nature across the seasons, in which case birds may be unable to ameliorate such a tradeoff through careful choices of diet.”